# Estimating instantaneous sea-ice dynamics from space using the bi-static radar measurements of Earth Explorer 10 candidate Harmony

Marcel Kleinherenbrink[1], Anton Korosov[2], Thomas Newman[3], Andreas Theodosiou[1], Alexander S. Komarov[4], Yuanhao Li[1], Gert Mulder[1], Pierre Rampal[5], Julienne Stroeve[3], and Paco Lopez-Dekker[1]

[1]Geoscience and Remote Sensing, Delft University of Technology, Delft, The Netherlands
[2]Nansen Environmental and Remote Sensing Center, Bergen, Norway
[3]Centre for Polar Observation and Modelling, University College London, London, United Kingdom
[4]Data Assimilation and Satellite Meteorology Research Section, Environment and Climate Change Canada, Ottawa, Canada
[5]Institut Géophysique de l'Environnement, CNRS, Grenoble, France

**Correspondence:** Marcel Kleinherenbrink (m.kleinherenbrink@tudelft.nl), Paco Lopez-Dekker (f.lopezdekker@tudelft.nl)

**Abstract.** This article describes the observation techniques and suggests processing methods to estimate dynamical sea-ice parameters from data of the Earth Explorer 10 candidate Harmony. The two Harmony satellites will fly in a reconfigurable formation with Sentinel-1D. Both will be equipped with a multi-angle thermal infra-red sensor and a passive radar receiver, which receives the reflected Sentinel-1D signals using two antennas. During the lifetime of the mission, two different formations will be flown. In the stereo formation, the Harmony satellites will fly approximately 300 km in front and behind Sentinel-1, which allows for the estimation of instantaneous sea-ice drift vectors. We demonstrate that the addition of instantaneous sea-ice drift estimates on top of the daily integrated values from feature tracking have benefits in terms of interpretation, sampling and resolution. The wide swath instantaneous drift observations of Harmony also help to put high-temporal resolution instantaneous bouy observations into a spatial context. Additionally, it allows for the extraction of deformation parameters, such as shear and divergence. As a result, Harmony's data will help to improve sea-ice statistics and parametrizations to constrain sea-ice models. In the cross-track interferometry (XTI) mode, Harmony's satellites will fly in close formation with an XTI baseline to be able to estimate surface elevations. This will allow for improved estimates of sea-ice volume, and also enables to retrieve full two-dimensional swell-wave spectra in sea-ice covered regions without any gap. In stereo formation, the line-of-sight diversity allows to infer swell properties in both directions using traditional velocity bunching approaches. In XTI mode, Harmony's phase differences are only sensitive to the ground-range direction swell. To fully recover two-dimensional swell-wave spectra, a synergy between XTI height spectra and intensity spectra is required. If selected, the Harmony mission will be launched in 2028.

## 1 Introduction

Sea ice plays a vital role in the climate system, reflecting sunlight and acting as an insulator between the ocean and the atmosphere. It also provides an important habitat for marine biota and serves as a platform for coastal populations (Krupnik,

I. et al., 2010; Dammann et al., 2019b). Reductions in sea-ice area and volume therefore have local as well as global impacts and require careful monitoring. Satellites play a key role in this monitoring, and have documented large-scale reductions in Arctic sea-ice extent year round since the late 1970s (Stroeve and Notz, 2018), and support climate model simulations that the Arctic will likely transition towards seasonally ice-free conditions before the middle of this century as a result of increases in

atmospheric greenhouse gases (Notz et al., 2020). In contrast, over the same time-period, Antarctic sea-ice extent has exhibited slight positive increases until 2014 when ice conditions started to reduce, highlighting the strong influence of natural climate variability in this region (Parkinson, 2019). While satellites have monitored large-scale sea ice extent for more than 40 years, sea-ice thickness on a large-scale was only sampled sporadically in the 2000s from ICESat (Lindsay and Schweiger, 2015) and is currently monitored using both CryoSat-2 (Ricker et al., 2017) and ICESat-2 (Kwok et al., 2019; Petty et al., 2020).

Nevertheless, this data has suggested an overall decline in Arctic sea-ice thickness and hence total ice volume (Ricker et al., 2014; Armitage and Ridout, 2015), while thickness trends in the Antarctic remain uncertain (Price et al., 2019).

Even though large-scale trends in extent and thickness are apparent, the processes governing regional sea-ice changes remain poorly observed. Most of the regional processes that cause changes in sea-ice volume are related to sea-ice drift. Small-scale volume changes are the direct effect of changes in the net drift of sea ice into an area, which is primarily driven by ocean

currents and wind stress (Dierking et al., 2017). Indirectly, sea-ice drift opens leads, which exposes the warm sea water to the colder atmosphere, and the formation of pressure ridges, which act like sails and keels for the atmosphere and ocean to exert forces upon. Observations of sea-ice drift are made from a variety of spaceborne instruments, like scatterometers, side-looking radars and optical sensors, as well as terrestrial observations from drifting bouys (Sumata et al., 2014; Long, 2017). Spaceborne Synthetic Aperture Radar (SAR) provides, with the use of methods like feature tracking (Korosov and Rampal, 2017), high-

resolution observations, while covering both poles with only a small gap. The feature-tracking approach allows for integrated values obtained between two satellite overpasses, at a sampling rate that depends on latitude, but is typically once per day. Daily integrated values however undersample the sea-ice dynamics and lead to an underestimation of the drift speeds (Haller et al., 2014; Dammann et al., 2019b). This is particularly true for breakup events, where the instantaneous velocities are several times larger than the daily averages (Karvonen, 2016). The application of feature tracking to estimate drift velocities in marginal

ice zones is also limited, due to the difficulty of tracking the more dynamic small ice floes. The second method to estimate sea-ice drift from SAR is to use the Doppler effect based on Doppler centroid anomalies, which provides an instantaneous one-dimensional estimate of the sea-ice drift (Kramer et al., 2015). One-dimensional quasi-instantaneous drifts with improved sensitivity have been estimated using Tandem-X data based on Along-Track Interferometry (ATI) (Dammann et al., 2019a). However, Tandem-X ATI observations have only limited availability. The latter two methods have the advantage that they also

perform well in the marginal ice zone.

Ardhuin et al. (2017) and Stopa et al. (2018) demonstrated the estimation of wave spectra using SAR in sea-ice covered regions. The approach exploits the phenomenon of velocity bunching, which is the displacement of scatterers in the azimuth direction of the SAR image as a consequence of the motions of swell waves in the range direction. Velocity bunching does not occur when waves travel in the range direction, so the two-dimensional swell-wave spectra are not strongly constrained from

55 a single pass. The spectra are further contaminated by sea-ice features e.g., ridges, that cover a similar spectrum as the swell

waves (Ardhuin et al., 2017). Some processes, like fracturing of floes are related to the penetration of long-wavelength ocean waves into the sea ice (Squire, 2018). As only a few limited studies have been performed to observe swell-wave dissipation in sea-ice-covered regions, this is currently poorly constrained in models. Using Sentinel-1 it is possible to estimate swell spectra over 5 km $\times$ 7 km regions based on a modulation of the amplitude in SAR images caused by swell waves. However, swell

waves that propagate across track are almost not visible as the dominant modulating mechanism, velocity bunching, is absent in this direction (Ardhuin et al., 2017).

The proposed bi-static Harmony SAR mission aims to provide new data for a wide range of applications, such as ocean-atmosphere interaction, hurricanes, solid Earth and land ice studies (López-Dekker et al., 2019). Harmony has also the capability to overcome several limitations of current sea-ice observing systems. The Earth Explorer 10 candidate mission consists of

two companion satellites (Concordia and Discordia), which will fly in formation with one of the Sentinel-1 satellites (from here on it is assumed to be Sentinel-1D). The satellites will carry a multi-angle thermal infrared sensor and two passive instruments that will receive signals from a scene illuminated by the radar of Sentinel-1D. In the stereo phase of the mission, the Harmony satellites are trailing and leading Sentinel-1D, which creates line-of-sight diversity. A two-channel receiver system onboard the satellites allows the radial velocity component of the surface to be retrieved, which in combination with the line-of-sight

diversity enables the estimation of instantaneous two-dimensional velocity fields, which in case of polar waters is sea-ice drift. The two-dimensional instantaneous drift observations of Harmony allow for a better interpretation of high temporal resolution observations from bouys and statistical validations of high-resolution sea-ice models. In cross-track interferometry (XTI) mission phases, the Harmony satellites fly in a close formation with a cross-track or radial baseline of several hundreds of meters. The observed phase differences between the two receivers are a result of surface topography, from which swell-wave properties

can be inferred. This is not possible with XTI using two monostatic systems, nor with repeat-pass interferometry, due to the decorrelation of the surface. In both XTI and stereo formation SAR spectra are obtained from multiple lines-of-sight which will improve the wave spectrum retrieval.

The objective behind this article is to describe the expected performance and the application of the Harmony mission for sea-ice dynamics i.e., sea-ice drift and waves. Harmony data also enables estimates of sea-ice topography, but this will not be

addressed in this paper. We will introduce the bi-static geometry and the observation concept and show how to derive sea-ice drift fields from the stereo configuration and wave spectra in both the stereo and XTI configurations. Harmony observations are modelled based upon sea-ice drift estimates from a sea-ice model and noise is added using electromagnetic models for the noise-equivalent sigma zero (NESZ) and the backscatter coefficient over sea-ice. We will discuss the performance in terms of accuracy, resolution and sampling. With dedicated filters and edge detection algorithms, it is further shown that the data from

Harmony is suited for the estimation of deformation parameters, like shear and divergence, at discontinuities. As a final step, we demonstrate how wave-spectra can be retrieved based on an end-to-end simulator and argue that Harmony is able to retrieve two-dimensional wave spectra in both formations.

## 1.1 Harmony's observation geometry

This paper does not attempt to detail the baselines over the orbit, but assumes realistic values wherever necessary to compute the performance of Harmony. A detailed overview of the observation geometry of Harmony will be discussed in a separate publication. Figure 1 shows the reconfigurable formation of Harmony and table 1 the relevant mission parameters. By several orbit manoeuvres Harmony-B, or Discordia, can be moved from the stereo to the XTI position, which typically takes one to two months. In the expected mission lifetime of five years, Harmony will fly approximately two years in the XTI formation and three years in the stereo formation.

As visible in figure 1, both satellites carry a passive radar instrument that receives the C-band echoes from the swath illuminated by Sentinel-1D at two (or three) phase centers, separated several meters in the along-track direction. The two phase centers can be used as a single receiver to get improved radiometric performance, or as two separate receivers to allow for short-baseline along-track interferometry (ATI). The phases differences of ATI are a result of range direction motion of the surface in the effective time separation between the antennas, which is related to the effective along-track separation of the antennas and the platform velocity. The effective along-track separation differs from the physical baseline between the two phase centers as the satellites have to be slanted to point the beams towards the illuminated swath. Since the stereo formation has two-lines of sight, the ground motion will be observed in two directions, allowing for the separation of azimuth and cross-track direction of sea-ice drift.

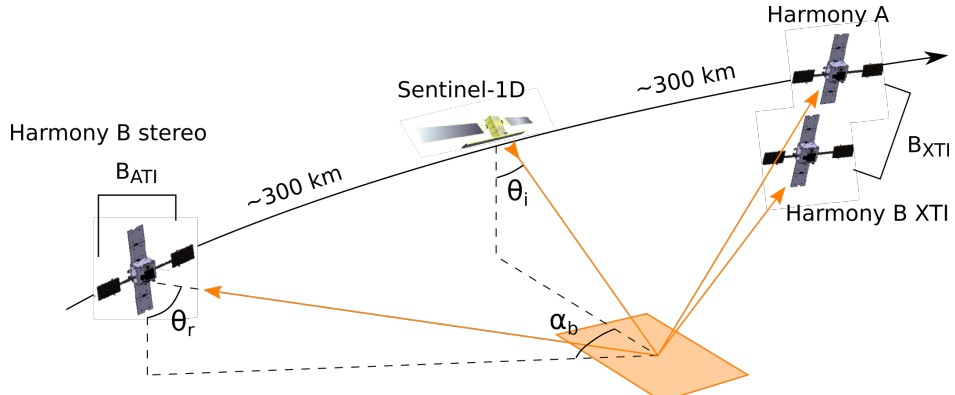

**Figure 1.** The reconfigurable configuration of the Harmony-Sentinel-1 constellation. In the stereo formation one of the Harmony satellites trails Sentinel-1 and the other heads Sentinel-1. In the XTI configuration, the Harmony satellites fly in close formation. Adapted from Kleinherenbrink et al. (2019).

In the close formation, inter-satellite interferometry can be applied, which makes the phase sensitive to motion if there is an effective along-track separation, or height, if there is an effective cross-track baseline. The range difference depends on the receiver positions only for XTI, so the effective XTI baseline to estimate elevation is in the plane of the received signal. The geometry of Harmony allows to minimize the inter-satellite ATI baseline, but variations over the orbit occur, which makes the phase sensitive to ground motion. To estimate topography, short-baseline ATI between the two phase centers on one satellite is

| parameter | symbol | value | unit |
|---|---|---|---|
| ATI baseline | $B_{ATI}$ | $\sim$9 | m |
| XTI baseline | $B_{XTI}$ | $\sim$100-1000 | m |
| wavelength | $\lambda_0$ | 5.6 | cm |
| platform velocity | $V_{sat}$ | 7 | km s$^{-1}$ |
| incident angle Sentinel-1 (IWS) | $\theta_i$ | $\sim$30-46 | ° |
| incident angle Harmony (IWS) | $\theta_r$ | $\sim$37-50 | ° |
| bistatic angle (IWS) | $\alpha_b$ | $\sim$23-35 | ° |
| instrument noise | NESZ | $\sim$-18–26 | dB |

**Table 1.** Relevant mission parameters. IWS refers to the interferometric wide swath mode considered in this study.

used to determine and correct for the ground-range motion. If high radiometric performance is required and both phase centers
are used to form a single antenna, methods like Doppler centroid anomaly estimation could be used for the ground-motion correction (Madsen, 1989). Doppler centroid estimation has an inferior accuracy with respect to short-baseline ATI and puts requirements on the surface i.e., strong variations in NRCS over the scene would introduce biases.

While the bi-static nature of Harmony provides the opportunity for estimating instantaneous two-dimensional velocities and instantaneous elevations, there are some observational consequences. ATI and XTI phase difference are a factor of two smaller
than in interferograms computed from overpasses of two monostatic systems, because there is only one transmitter and two receivers. Additionally, the position of the receivers causes a rotation of the observed polarization. The coherent geometrical polarization change is predictable and can be anticipated, either in the design or in the processing of the data. The non-coherent part reflects the change of polarization in different lines-of-sight, and will contain additional information about the geometry of the surface, like directional ocean waves. Finally, as will be discussed in section 3, waves modulate the amplitudes depending
on their orientation with respect to the transmitter-receiver system. This will make Harmony sensitive to all directions of swell.

## 2  Methods

The performance of Harmony's potential for sea-ice-drift observations is analysed with the use of the neXtSIM sea-ice model. From the model input, the noise-free multilooked ATI phases are computed and noise is added using electromagnetic models for the backscatter and the NESZ. After addition of noise, the process is reversed to get an estimate of Harmony's surface velocity
estimates. Filtering and edge detection methods are suggested to de-noise the velocity fields, while keeping discontinuities intact. We additionally briefly introduce an OceanSAR end-to-end model, which is used to simulate radar signals from swell waves in sea-ice-covered waters and demonstrate the ability of Harmony to recover swell signals based on both XTI and stereo phase and amplitude spectra.

## 2.1 Sea-ice model

If launched, Harmony will be the first mission that is able to measure instantaneous displacement patterns for sea-ice similarly to what is commonly measured for the Earth crust, and subsequently to reveal new and crucial information about the sea-ice fracturing dynamics itself. To be able to simulate Harmony's performance, we take our input sea-ice dynamics from the latest version of the Lagrangian neXtSIM model (Rampal et al., 2019). The Maxwell-Elasto-Brittle rheological model for sea-ice included in neXtSIM is capable to capture large-scale sea-ice drift and deformation statistics as observed from satellite over a wide range of spatial and temporal scales (1—1000km and day—month, respectively) (Rampal et al., 2016, 2019). A pan-Arctic simulation with a time-step of 200 seconds and a horizontal resolution of approximately 3.5 km x 3.5 km has been performed. The stereographic coordinates and velocities have been converted to ground-projected radar coordinates and velocities, and then interpolated to the resolution set by the experiments in the later sections. A median filter is applied to remove interpolation effects by the conversion from a triangular to a Cartesian grid, while keeping the discontinuities intact.

## 2.2 Phase measurements

The primary focus of the paper is the estimation of sea-ice drift for which Harmony applies short-baseline ATI between two phase centers on one satellite. The radial velocity of the surface can be measured by two acquisitions of the surface separated in time. ATI makes use of the phase difference caused by a time offset between the two coinciding Doppler centroids i.e., after processing the signals received at the two antennas can be considered as two acquisitions of the same scene separated by a time difference $\Delta t$. In case of a monostatic systems, this is directly related to the along-track baseline $B_{ATI}$, such that the time difference is given as

$$\Delta t_m = \frac{B_{ATI}}{V_{sat}}, \tag{1}$$

where $V_{sat}$ is the platform velocity. In a bi-static system, like Harmony, where there is one transmitter and two receivers, the Doppler Centroid moves only half the baseline distance, which results in an effective time difference of

$$\Delta t_b = \frac{B_{ATI}}{2V_{sat}}. \tag{2}$$

The Doppler shift measured by Harmony is, in contrast to standard monostatic systems, not only sensitive to across-track surface velocity $v$, but also to its along-track component $u$. Using the orientation and geometry described in the previous section, the Doppler shift is given by Kleinherenbrink et al. (2019)

$$f_{D,\pm} = \pm \frac{\sin(\theta_r)\sin(\alpha_b)}{\lambda_0} \cdot u - \frac{\sin(\theta_i) + \sin(\theta_r)\cos(\alpha_b)}{\lambda_0} \cdot v, \tag{3}$$

where $\pm$ is negative for Discordia and is positive for Concordia and $\lambda_0$ is the wavelength of the carrier frequency. The incident angles for transmitter $\theta_i$ and receiver $\theta_r$ in this equation are differ a few degrees and depend on the ground-projected bistatic angle $\alpha_b$. For the geometric interpretation of these angles we refer to figure 1. The phase difference caused by surface velocities is computed as Duque et al. (2010)

$$\Phi_{ATI,\pm} = 2\pi \cdot f_{D,\pm} \cdot \Delta t_b. \tag{4}$$

From equation 3 it follows that the along-track velocity can be reconstructed by subtracting the phase differences of Concordia and Discordia and the cross-track velocity by a summation under the assumption that the distance between Sentinel-1 and the two Harmony satellites is almost equal. This yields the most important equations for Harmony's sea-ice drift retrieval

$$\hat{u} = \frac{\lambda_0}{2\pi\Delta t_b} \frac{\hat{\Phi}_{ATI,+} - \hat{\Phi}_{ATI,-}}{2\sin(\theta_r)\sin(\alpha_b)}$$

$$\hat{v} = -\frac{\lambda_0}{2\pi\Delta t_b} \frac{\hat{\Phi}_{ATI,+} + \hat{\Phi}_{ATI,-}}{2(\sin(\theta_i) + \sin(\theta_r)\cos(\alpha_b))} \ , \tag{5}$$

where $\hat{\Phi}_{ATI,\pm}$ are the observed phases. In case the geometry of the constellation is not symmetric, such that the bistatic angle
$\alpha_b$ is different for the two satellites, a more elaborate equation for inversion is required.

Height sensitivity in SAR is achieved by using acquisitions of the same scene separated by a cross-track distance. In side-looking monostatic systems, the two-way range difference between two passes can directly be coupled to surface elevation. For Harmony the same is true, except that it only depends on a single-way range difference as there is only one transmitter and two receivers in a cross-track formation. Note that in case of a bi-static system, the effective XTI baseline $B_{XTI}$ is also not in
the cross-track direction, but along the line-of-sight of the receivers. The height of ambiguity is given by

$$H_a = -\frac{\lambda_0 r_1 \sin(\theta_r)}{B_{XTI} \cdot \cos(\theta_r - \alpha)} \tag{6}$$

with $r_1$ the range toward one of the receivers and $\alpha$ is the slope angle of the XTI baseline. Then the interferometric phase difference caused by topography $h$ is computed with

$$\Phi_{XTI} = 2\pi \frac{h}{H_a}. \tag{7}$$

## 2.3 Noise modelling and bi-static backscatter

Realistic values for retrieved velocities and elevations are obtained by adding noise to the forward-modelled interferometric phases. The interferometric phase noise is related to the coherence $\gamma$ and is computed as (Dierking et al., 2017; Rosen et al., 2000)

$$\sigma_\phi = \sqrt{\frac{1 - \gamma^2}{2N\gamma^2}}, \tag{8}$$

where $N$ is the number of independent looks. Using the equations from section 2.2, the phase noise is converted to velocity and elevation noise. For the computation of the phase noise, the coherence $\gamma$ is modelled, which is by a multiplication of several different parts, such that

$$\gamma = \gamma_{sys} \cdot \gamma_{XT} \cdot \gamma_{AT} \cdot \gamma_{vol}, \tag{9}$$

where $\gamma_{sys}$, $\gamma_{XT}$ and $\gamma_{AT}$, respresent the system coherence and the range and along-track baseline decorrelation, respectively.
The term $\gamma_{vol}$ related to volume decorrelation is set to 1 if short-baseline ATI is considered as the baseline is only a few meters. For XTI mode, $\gamma_{vol}$ reduces and it depends on the ice properties.

In the close formation, inter-satellite phase differences are computed, so that the along-track and cross-track baselines cause significant changes to the spectra. During Stereo formation flying, where we use a separation of a few meters between the phase centers, it is possible to ignore baseline decorrelation. In the XTI mode the decorrelation due to the XTI baseline decorrelation cannot be ignored as the satellites are separated by $> 100$m, while using a helix formation in the slanted geometry of Harmony, the along-track baseline will be only a fraction of the cross-track baseline over a large fraction of the orbit. The decorrelation caused by these spectral shifts is related to the critical baselines in both directions. In the range direction the critical baseline is given as

$$B_{XTI,crit} = \frac{\lambda_0 a}{\Lambda_{ra} \cos^2(\theta)},$$

(10)

whereas

$$B_{ATI,crit} = \frac{\lambda_0 a \cos(\theta)}{\Lambda_{az} \sin(\theta)}$$

(11)

is the along-track critical baseline, with $a$ the satellite altitude, and $\Lambda_{az}$ and $\Lambda_{ra}$ the resolution in the azimuth and range directions, respectively. The combination of both baselines, $B_{ATI}$ and $B_{XTI}$, yields a coherence of (Dierking et al., 2017)

$$\gamma_{XT} = 1 - \frac{B_{XTI}}{B_{XTI,crit}}$$

$$\gamma_{AT} = 1 - \frac{B_{ATI}}{B_{ATI,crit}}.$$

(12)

The system coherence depends on the Signal-to-Noise Ratio (SNR), which in itself depends on the backscatter coefficients $\sigma_0$ and the Noise Equivalent Sigma Zero (NESZ), such that

$$SNR = \frac{\sigma_0}{\text{NESZ}}.$$

(13)

Then the corresponding coherence is given by

$$\gamma_{sys} = (1 + \frac{1}{\text{SNR}})^{-1}$$

(14)

The NESZ depends on the system parameters and the antenna gains of both Sentinel-1 and Harmony. If both phase centers onboard the Harmony satellites are used as a single antenna, a better radiometric performance is achieved. In case of short-baseline ATI, both phase centers are used separately, causing the beam to widen and the NESZ to increase. The NESZ can be computed using

$$\text{NESZ} = \frac{E_n}{E_s},$$

(15)

with

$$E_n = k T_{sys} BW,$$

(16)

where BW is the bandwidth, $T_{sys}$ is the noise temperature and $k$ is Boltzmann's constant. The received signal energy as a function of the transmitted energy $E_p$ is given by the radar equation

$$E_s = E_p(p_{2w}\frac{\lambda_0}{(4\pi)^{1.5}r_1^2}\sqrt{A_{res}}G_{0,tx}G_{0,rx})^2, \tag{17}$$

with $p_{2w}$ the two-way antenna pattern $A_{res}$ the resolution and $G_{0,tx}$ and $G_{0,rx}$ the maximum transmitter and receiver gains, respectively. The antenna patterns are computed for two 0.7 m × 3.2 m areas separated by approximately 9 m, which are approximate dimensions set by preliminary feasibility studies from industry. For the single phase center Harmony receiver, the NESZ varies over the swath (figure 2) and has typical values of -22 dB.

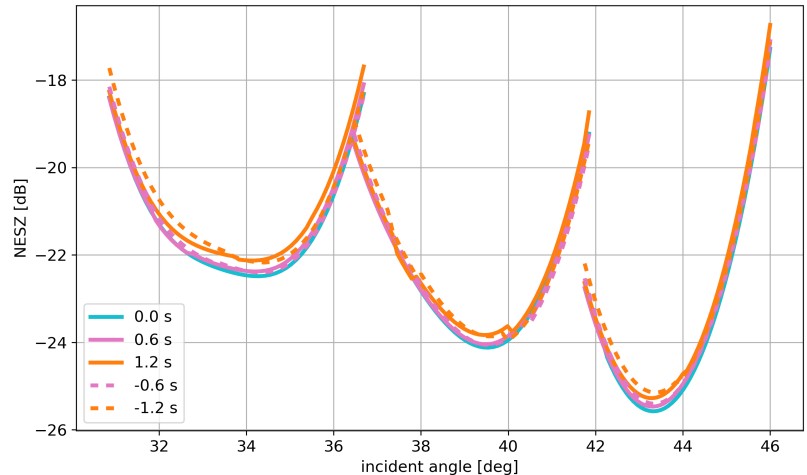

**Figure 2.** The estimated NESZ for Harmony for three azimuth locations: at zero Doppler (0.0 s),at about 5 km from the zero Doppler (0.6 s) and at about 10 km from zero Doppler (1.2 s). The subswath edges of Sentinel-1's IW mode are visible by the discontinuities.

Scattering characteristics of sea ice depends on many factors including antenna orientation, surface and subsurface rough-
220 ness, and dielectric profiles of snow and ice. Dierking et al. (2017) summarized the range of backscattering coefficients for various types of sea ice based on previous studies. Many analyses of the backscatter coefficient have been performed over various types of ice (Kwok and Cunningham, 1994; Dierking, 2010; Ulaby and Long, 2015). C-band backscattering coefficients typically ranged from -23 to -8 dB for monostatic systems. Bi-static estimates in C-band have not been made so far, so we rely on a two-layer implementation (snow and sea ice) of the model of Komarov et al. (2014) and Komarov et al. (2015). The
225 backscatter coefficients at the interfaces for polarization $xx$ are summed to get total backscatter coefficient, such that:

$$\sigma_{xx} = \sigma_{xx,air-snow} + \sigma_{xx,snow-ice}. \tag{18}$$

The model for the two-layer snow-ice system is further elaborated in the appendix. Note that the implemented snow-ice model assumes two parallel interfaces, which is in reality not always true. This model allows us, however, to provide performance estimates for a typical backscattering scenario.

## 2.4 Post-processing

To be able to derive accurate deformation parameters near discontinuities (e.g., shear and pressure zones), averaging should be applied parallel and not perdendicular to these features. However, the identification of these structures is non-trivial, because even after some multilooking the noise might be large enough to cover the discontinuities. To demonstrate that it is indeed possible to identify gradients, we designed a two-step solution, which comprises of a variant of the Wiener filter and an edge detection method. The adaptive Wiener filter ensures that the scenes become interpretable and that the noise is suppressed, while maintaining the sharp gradients. With the edge detection method, the gradients in velocity are detected, so that consecutively suitable averaging methods can be applied to compute the instantaneous pressure and shear. Note that the method described here is only one of the possible processing flows. Other, more advanced methods, can be developed once Harmony is delivering data. The settings and thresholds used in the described method can also be varied, depending on the application.

As the backscatter coefficient and NESZ are known, an estimate can be made of the velocity noise $\sigma_n$. Under the assumption that 2 km $\times$ 2 km multi-looking is applied, speckle is suppressed and the velocity noise is considered to be Gaussian. Using a two-dimensional discrete Fourier transform of the along- or across-track velocity field $V_{u,v}$, the power spectral density is estimated as

$$\Phi_{u,v} = \frac{|V_{u,v}|^2}{N \cdot M}, \tag{19}$$

where $N$ and $M$ are the dimensions of the velocity field. Unfortunately, the spectrum of the true velocity field are not known, so therefore we make an estimate, such that:

$$\Phi_{(u,v),true} = \Phi_{u,v} - s\sigma_n^2, \tag{20}$$

where $s$ is a scale factor for the noise. Negative values for $\Phi_{(u,v),true}$ cannot exist, so they are set to 0. The completely removes the signal at some frequencies where the signal is close or below the noise level, causing a small bias. Our adaptive Wiener filter is then given as:

$$W = \frac{\Phi_{(u,v),true}}{\Phi_{u,v}}, \tag{21}$$

so that our filtered velocity fields in the frequency domain read

$$V_{(u,v),f} = W V_{u,v}. \tag{22}$$

Even though the noise is suppressed by the Wiener filter, we find that standard edge detection methods, such as the Canny edge detector, are not suitable. Therefore, we opt for an alternative involving multiple neighbouring pixels. To detect potential edges in the azimuth and ground-range direction, we convolve the Wiener filtered fields with edge-detection filters of size N $\times$

2N+1 and 2N+1 × N of which one part contains the values $\frac{-1}{N^2}$ and the other $\frac{1}{N^2}$. The size of N is currently set to 15 pixels, but might be varied depending on the noise in the scene and user requirements (i.e. false detections, accuracy and others). Discontinuities are represented by peaks in the filtered velocity fields and are detected by an algorithm that scans images line-by-line. These operations are applied to the $u$ and $v$ velocity fields in both directions.

To fix broken edges, a binary image of potential edge locations is created and dilated. Regions identified as potential edges are kept if they have more than 50 pixels connected together in the dilated binary images, which implies that the discontinuities have to be at least several tens of kilometers in length. To sharpen the edges again, the binary image is eroded. For every remaining edge location, the edge-detection filter with size N=5 is applied to compute the gradients. Using the gradients, we determine the absolute value of the shear as

$$\tau = \sqrt{(\frac{dv}{dx})^2 + (\frac{du}{dy})^2},\tag{23}$$

and the divergence as

$$\nabla = \frac{du}{dx} + \frac{dv}{dy}.\tag{24}$$

The shear and divergence are expressed in cm s$^{-1}$ and not normalized by dividing by the pixel resolution as the shearing and divergence along edges typically occurs over a distance smaller than the resolution.

## 2.5 Wave spectra

The software package OceanSAR (https://github.com/pakodekker/oceansar) is used to demonstrate Harmony's ability to measure swell waves in sea-ice-covered regions. Harmony data can be used in two ways to extract wave spectra. If Harmony flies in the XTI configuration, the phase differences between the two satellite systems are a 'direct' measure for the height. In both the XTI and Stereo formations, the image distortions due to velocity bunching provides a way of inferring wave information from intensity spectra. To the first order, a bi-static observation can be modelled as a mono-static system located at the midpoint between the transmitter and receiver. In this paper, we consider a system of one transmitter and two receivers (XTI mode), so that we have a monostatic and a bi-static system of which the bi-static receiver is located across track. This also implies that both systems have a common zero-Doppler, which makes the interpretation easier. Using OceanSAR, the RAW signals are modelled using scatterers at a 2 m resolution Lagrangian grid i.e., the swell waves cause three-dimensional changes to the surface during the burst. As the small waves are quickly damped by the sea ice (Stopa et al., 2018), the surface is assumed to be correlated over the burst length. The average SNR is set to approximately 5 dB and the bi-static baseline to 1000 m.

The data is then focused, and has a comparable resolution as the Sentinel-1's IW mode. From the single-look complex images intensities and interferometric phases are computed. These are multilooked using a Hanning window with a size of 12 × 4, which yields a resolution of about 65 m × 65 m, since the simulated resolution is approximately 16 meter in azimuth and 5 m in the ground-range direction. Using the equation for the height of ambiguity in section 2.2, the surface elevation is computed from the phases. Spectra are estimated using Bartlett's method (Bartlett, 1950), using eight non-overlapping patches

over the 4 km × 4 km grid. Finally, by squaring the mean periodogram and scaling it by the number of samples the amplitude spectrum is computed.

## 3 Discussion of the results

The measurements of Harmony and its characteristics are addressed in four sections. The first section discusses the expected SNR and the overall performance of the system for surface velocity vector estimation from Harmony data. Then we address the coverage in terms of the number of passes and the expected accuracy in each pass. The third section focuses on strategies to reveal structure and estimate accurate gradients. Finally, we have a separate discussion on the estimation of wave spectra from intensity images and XTI elevations.

### 3.1 Performance metrics

An estimate of the bi-static NRCS is required to compute the performance of Harmony. As mentioned before, based on other studies the backscatter coefficient of sea ice typically ranges from -23 to -8 dB, depending on the surface properties and the incidence angle. These numbers are used to constrain the snow and ice properties, which are input for the model. The backscatter is then computed as a function of bi-static angle and incidence angle (figure 3). In the right panel, the monostatic case is plotted for reference. The monostatic backscatter shows a typical decay with incidence angle, which is comparable to the patterns observed in Sentinel-1 and the study of Komarov et al. (2015). Volume scattering is often small for sea ice (Komarov et al., 2015) and is not modelled, hence the HV component is nearly zero. Therefore, the coherent sum of the HH and HV backscatter is approximately equal to the magnitude of HH. Note that volume scattering is not always negligible, for example over deformation features or multiyear ice (Scheuchl et al., 2005; Shokr, 2009).

The middle panel shows that a similar pattern is visible for the coherent sum of the backscatter in the bi-static case. However, the ratio between the HH and HV have changed as a consequence of the change in geometry. This is further illustrated by the left panel, where it is shown that the ratio HV-HH increases with increasing ground-projected bi-static angle. Note that towards the end of the swath the incidence angle increases, while the bi-static angle decreases, which causes the HV component to be more prominent at the near-field than at the far-field. The coherent sum of the backscatter only decreases slightly with increasing bi-static distance, hence the descrease in SNR is small. This result depends on the surface properties, but is fairly robust.

Under the assumption that the sum of the HH- and HV-polarized backscatter for the bi-static systems is comparable to that of a monostatic system, the performance of Harmony is computed for a range of SNRs (figure 4). At the considered 1 km × 1 km multilook (left panel) usable velocity estimates in both directions are expected at a SNR of 0 dB in Sentinel-1D's IW mode, because typical ice-drift velocities are below 0.5 m s$^{-1}$ (Scheiber et al., 2011; Dierking et al., 2017; Dammann et al., 2019b). In the worst-case scenario, at an SNR of -5 dB, a resolution of 2 km is sufficient to extract information about the fastest ice floes using Sentinel-1D's IW mode. For a typical situation with 5 dB SNR a substantial fraction of the sea-ice drift is resolved at 1 km x 1 km resolution. The accuracy changes over the swath, which is caused by the aforementioned variation in incidence

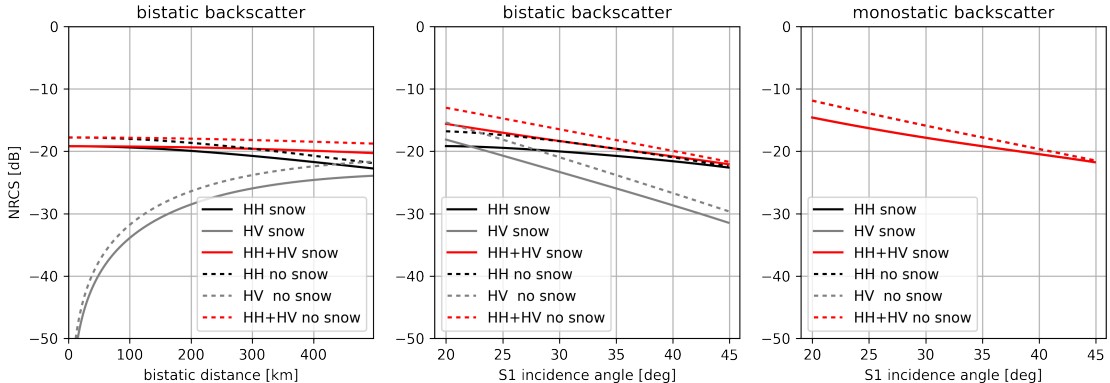

**Figure 3.** Plots of the modelled NRCS for the HH and HV polarizations as a function of bi-static distance (left) and the Sentinel-1 incidence angle (middle: bi-static case, right: monostatic case), while keeping along-track distance between Harmony and Sentinel-1 at 300 km. The solid lines correspond to a snow-covered ice surface and the dashed lines a snow-free scene. The dielectric constants are set to $1.6 + 0.07i$ and $3.65 + 0.38i$ for snow and ice, the surface roughness to 0.3 cm and the correlation length to 1.5 cm for both interfaces. The snow thickness is set to 4 cm. These values are close to a case study in Komarov et al. (2015).

and bi-static angles (right panel). Under the assumption of a constant SNR over the swath, this leads to an improved cross-track accuracy and a slightly decreased along-track accuracy. To achieve similar accuracy at least 4 km × 4 km of averaging is required when operating Sentinel-1D in the EW mode (middle panel), which is the current operating mode over most of the polar areas. A change of operational mode will therefore be beneficial for the retrieval of instantaneous drift.

## 3.2 Coverage and sampling

The wide swath of Sentinel-1D's EW mode was requested by the sea-ice community to increase the sampling rate for feature tracking algorithms to estimate sea-ice velocity. For the operation of Harmony, Sentinel-1D's IW mode is preferred, because of the enhanced number of independent looks, which allows to suppress the noise of velocity estimates. As the number of SAR missions increases and likely three Sentinel-1 satellites will be operated at the moment of launch, a request can be made to operate Sentinel-1D in IW mode over at least part of the sea ice. Therefore we will discuss both the coverage in the IW and the EW mode.

Figure 5 shows the estimated velocity from Harmony based on the input data from the neXtSIM model. The swath of the currently operated EW mode is about 400 km, which is about 140 km wider than the IW mode. The EW and IW modes have five and three subswaths, respectively, in which the noise increases towards the edges as a consequence of the antenna pattern. In reality the subswaths overlap partly, so that the noise can be reduced by approximately $\sqrt{2}$ at the edges. For the first subswath of the EW mode the antenna beamwidth should be widened to limit the increase in noise near the edges.

The wider swath of the EW mode comes at the cost of resolution, which decreases the number of independent looks for averaging. The reduced number leads to a worse precision of the velocity measurements, which is more-or-less a factor of $\sqrt{8}$

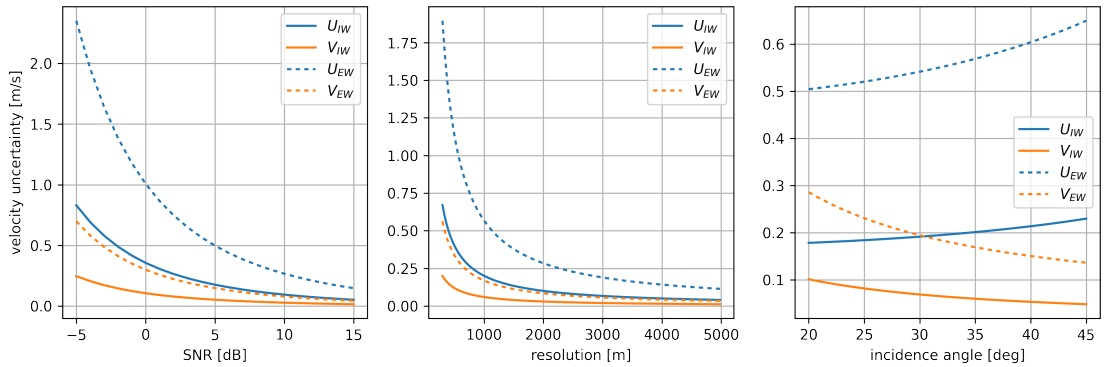

**Figure 4.** Modelled velocity uncertainties based on a flat Earth approximation. U and V represent the along- and across-track velocity uncertainties, respectively. The solid lines represent the performance for the IW mode, while the dashed line represent the performance for the EW mode. In the left panel the resolution is kept constant at 1 km × 1 km and in the middle panel the SNR is kept constant at 4 dB. The baseline between the two phase centers is set to 9 m and the incidence angle of Sentinel-1 to 35°. The distance between the Harmony satellites and Sentinel-1 is kept constant at 300 km. In the right panel, all of the previously mentioned settings are used to show the variations over the incidence angle.

(see previous section). This greatly reduces the ability to detect individual floes and discontinuities in the velocity field. The change of Sentinel-1D to the IW mode, which we opt for, comes at the cost of the number of passes (figure 6). Above 74°
latitude, the maximum number of passes in EW mode over any point is above twelve times per repeat-orbit, or once per day for a single Sentinel-1 satellite. By switching to the IW mode, this number will drop to about once per 1.5 day at the same latitude. Above 80° the number of once per day is still reached. For the sea ice surrounding Antarctica, which is located at lower latitudes, the sampling rate will be typically around once per two days. The expected number of passes is likely lower than the maximum as the number of observations is limited by the duty-cycle of Sentinel-1. Note that several other satellites
will be operated in C- and L-band in 2028, such as Sentinel-1, the RADARSAT constellation and ROSE-L, which ensure a sampling rate of better than once per day for feature tracking.

The great gain of having Harmony in stereo mode over sea ice is the ability to estimate instantaneous velocities, rather than only daily integrated values (figure 7). As shown in the figure, the velocity fields tend to change on sub-daily timescales, which leads to aliasing when only feature tracking is applied. The differences between daily and instantaneous observations
becomes even more apparent if the deformation is computed from the velocity fields. As a result the total energy transfer is underestimated in current daily-integrated velocities as also discussed in other studies (Dierking et al., 2017; Dammann et al., 2019b). The differences between daily integrated and instantaneous velocity are linked to short-term events, like the opening and closing of leads, floe collision and break-up. Harmony data allows to compute enhanced statistics on these type of events, which is beneficial for parametrization and calibration of sea-ice models. The addition of instantaneous velocities at two epochs
will also enhance the interpretation of daily integrated velocities and puts in-situ bouy observations into context.

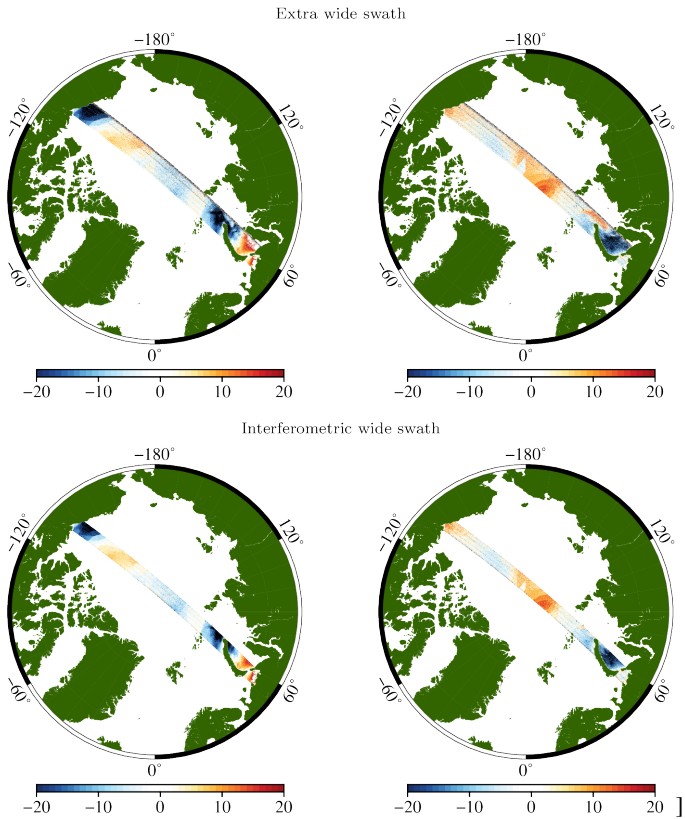

**Figure 5.** Modelled sea-ice velocity (cm s$^{-1}$) observations for a single pass of Harmony in the East (left) and North (right) directions for the EW and IW modes based on the 200 s integrated velocity fields of the neXtSIM model. The swath of the EW is clearly wider, which increases the temporal sampling at the costs of resolution.

Finally, in stereo formation, as long as the surface NRCS is sufficient (higher than $\sim -25$ dB), Harmony will be able to provide two-dimensional velocity estimates over any surface. This implies that velocity estimates can be obtained in the marginal ice zone over small and highly dynamic floes, where traditional methods as feature tracking have problems. In occasions when the water surface is exposed to strong winds, it is even possible to determine the velocity in leads, as already

demonstrated with Tandem-X (Dammann et al., 2019b). The separation of ocean surface current and sea-ice drift gives the opportunity to study the coupling between both.

### 3.3   Sea-ice drift and deformation

Figure 8 shows the estimated velocity fields from Harmony at 2 km $\times$ 2 km multilooking based on the neXtSIM model input. The figure demonstrates the result of the inferior along-track accuracy as discussed in section 3.1. Furthermore, the effect of

the antenna pattern is visible, because the noise increases near the edges of the subswaths. Without filtering, the velocity fields are difficult to interpret and to process.

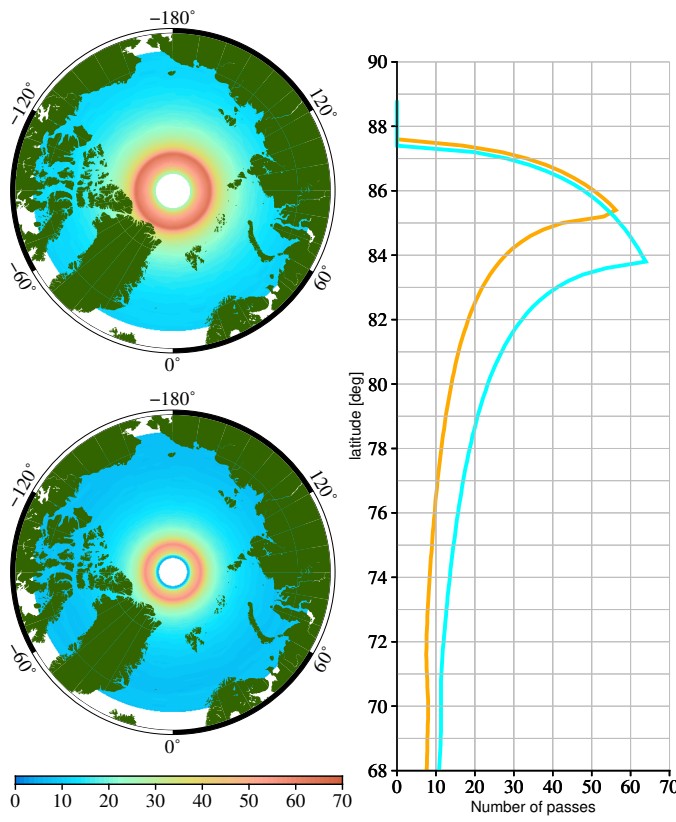

**Figure 6.** Maximum number of acquisitions within the 12-day repeat period for the EW mode (top, light-blue) and the IW mode (bottom, orange) as a function of latitude and geographically.

An adaptive Wiener filter is applied to reduce the noise, while it keeps the edges intact. Low-pass filters are less suitable, because edges require directional high-frequency harmonics, which would be suppressed. With a Wiener filter some higher frequencies are retained, such that the resolution effectively varies with direction. Rigorous changes in deformation patterns within a scene might lead to inferior results. It is therefore recommended to limit the size of the scene over which the adaptive Wiener filter is applied. For this particular scene, application of the Wiener filter reduces the root-mean-square of differences between the neXtSIM velocity fields and the filtered velocities from about 10 cm s$^{-1}$ to 2 cm s$^{-1}$ and from 4 cm s$^{-1}$ to less than 1 cm s$^{-1}$ for the azimuth and ground-range directions, respectively. The histograms in figure 8 show that Harmony is able to accurately capture the distribution of the velocities in both directions, as the differences with input fields are minimal.

We also derive the structure and deformation patterns of the sea ice, which would not be possible with feature tracking. Using edge detection methods for the velocity fields, we can identify separate regions (third and sixth panel of figure 8) and compute the shear and divergence at the edges of the floes (figure 9). This allows to extract statistics on the structure of sea ice, for example the floe size, even if there is no lead in between the floes. The estimated divergence at the edges indicates locations where leads open and close, and also identify locations where ridges might form as a consequence of collisions. Shear

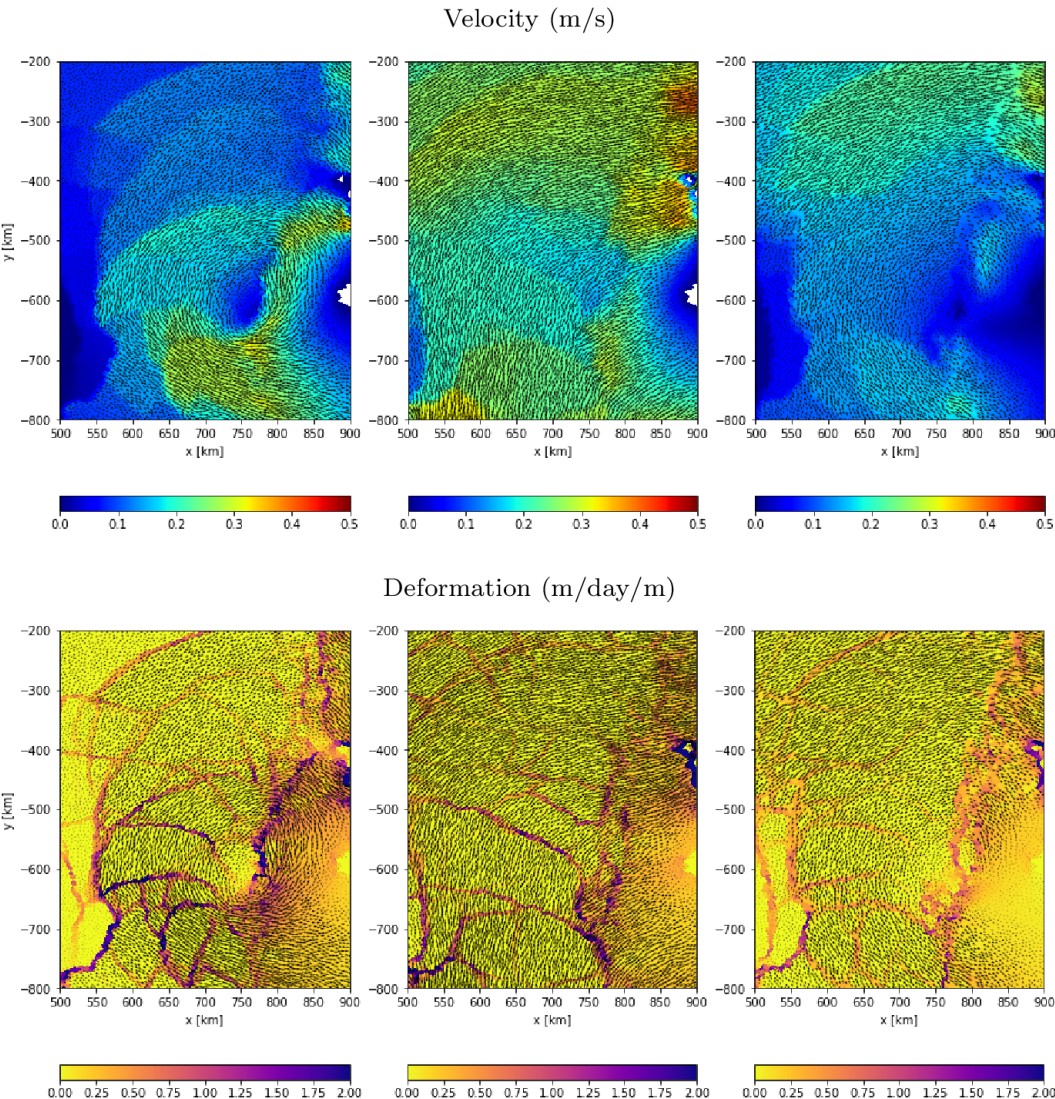

**Figure 7.** Instantaneous sea-ice velocities at two epochs separated by a day (top-left and top-middle) and daily integrated velocities (top-right) as estimated from the neXtSIM sea-ice model. In the bottom the associated deformation.

estimates are useful to determine sea-ice stress estimates and both shear and divergence are a source of information for local wind and ocean drag on sea ice. Statistics on these processes cannot be obtained with any operation mission currently flown.

The top histogram in figure 9 shows that the estimated shear at the boundaries follows a Rayleigh distribution. The absence of small shear is contributed to the limit of what we are able to extract with Harmony. Harmony, in combination with the proposed method is only able to reveal shear larger than $\sim$0.25 cm s$^{-1}$, which might be further enhanced by other more suitable signal processing strategies once it flies. Secondly, the shear is inevitably a bit underestimated due to the applied filter

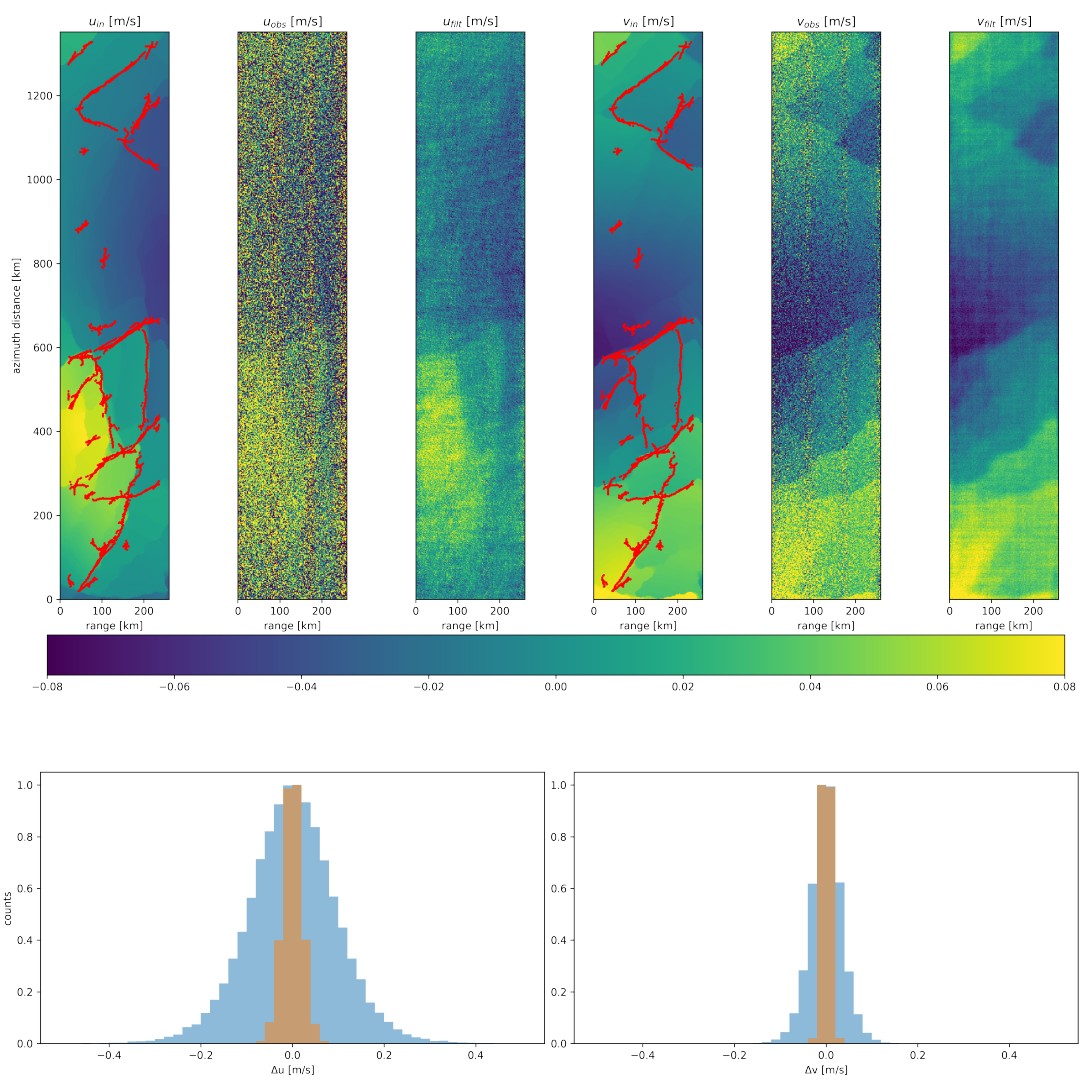

**Figure 8.** Azimuth and ground-range direction velocities retrieved by Harmony before and after application of the adaptive Wiener filter. The input data is also given with in green the detected discontinuities. The histograms on the right show the differences between the input and retrieved velocities before and after Wiener filtering.

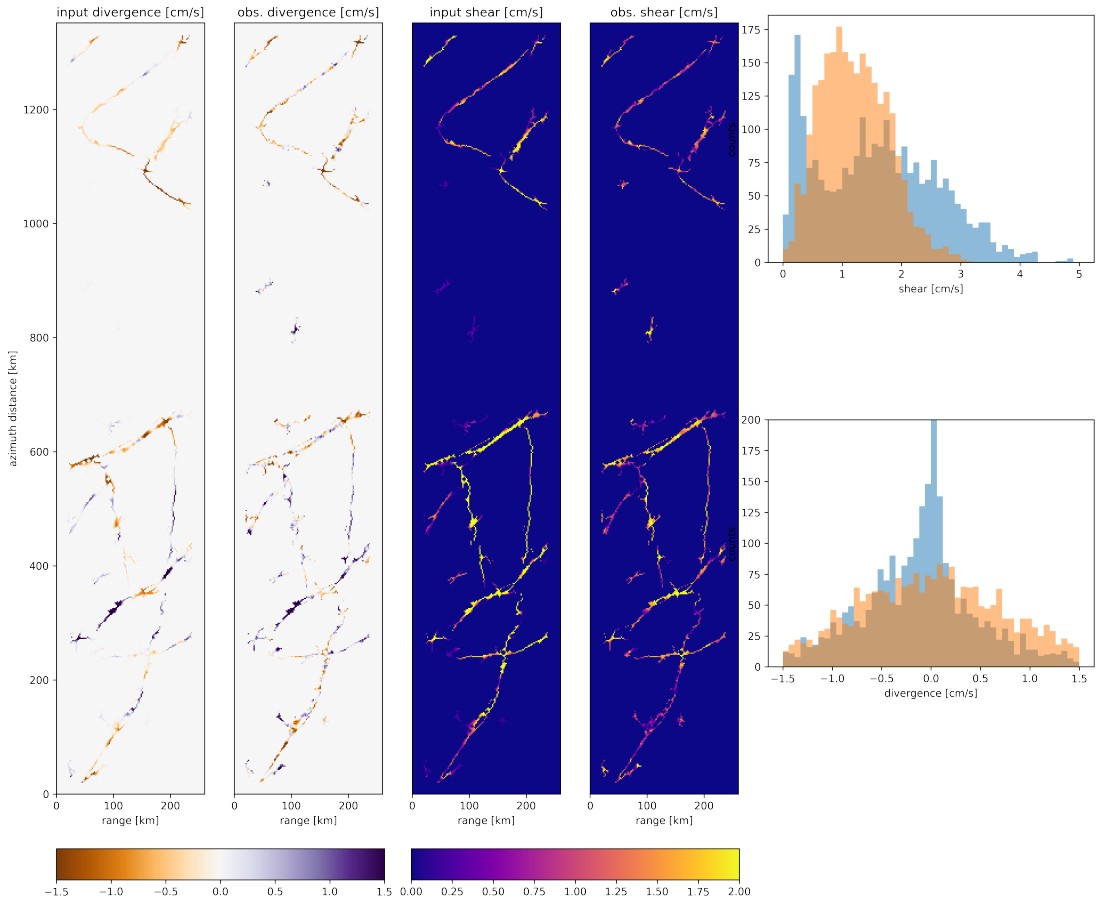

**Figure 9.** Estimated divergence and shear at the discontinuities from the input data and from the Wiener filtered data. The histograms show the distributions of the estimated divergence and shear over the same swath. The applied filter has an effect on the shear and divergence magnitude as particularly high-frequency signals are suppressed, which smooths the edges.

as it inevitably smooths the edges a bit. Less filtering decreases the underestimation, but introduces more noise in velocity estimates, which also has a consequence for the distribution. Following the equation for the divergence in sect 2.4, a higher noise in the gradients causes it also to be less likely for small divergence to be observed. It is therefore recommended that tailored filters should be applied, depending on the applications and whether divergence or shear is considered, which can for 390    example be done by scaling the noise parameter in the adaptive Wiener filter (sect 2.4).

Note that the results shown in this paper are an estimate of the performance of Harmony for sea ice based on the currently set system parameters, which likely will change before the launch of the mission. The assumed sea-ice backscatter in reality

also varies geographically, so the detection of discontinuities might be slightly better or worse depending on the local sea-ice properties. The suggested approach to filter the sea-ice drift estimates and estimate discontinuities might not be suitable for each type of sea ice and more advanced tailored solutions will likely perform better.

## 3.4 Wave spectra

Swell waves propagating in the azimuth direction modulate the amplitudes through velocity bunching, which is an artefact of focusing SAR images over non-stationary targets with harmonic behavior (Hasselmann and Hasselmann, 1991). As scatterers from different phases of the swell wave have different line-of-sight velocities, they introduce small Doppler shifts that cause clustering of scatterers in azimuth. The resulting surface motion to intensity modulation transfer-function is well known and is therefore already used to infer swell properties based on intensity spectra from Sentinel-1 data. Velocity bunching does not occur when swell waves propagate in the ground-range direction and are therefore weakly constrained with a monostatic SAR system alone. For a bi-static system the ground-range direction is approximately aligned with the midpoint between the transmitter and receiver, referred to as the equivalent monostatic geometry. Since the proposed Harmony mission uses two receivers and Sentinel-1D itself acts as one, in Stereo mode there are three lines-of-sight, each separated by approximately 16° (half the bi-static angle). This allows to infer swell properties with better sensitivity, especially near the range direction, than a monostatic system. When orbiting in the XTI formation, one line-of-sight is lost, such that we lose sensitivity to swell in the directions between the effective ground-range directions (-16° to 0°) of Harmony XTI and Sentinel-1D. Fortunately, the XTI phase differences also allow us to infer wave spectra, which are sensitive to the effective range direction of the bistatic system. An angle of zero in our simulations is approximately equivalent to waves propagating in the ground-range direction of a monostatic equivalent between the two Harmony satellites, or approximately -16°. Effectively, the XTI formation has therefore three lines-of-sight: intensity from Sentinel-1D only and from the bistatic system of Harmony and Sentinel-1 and phase difference from the XTI formation. A combination between intensity and elevation spectra would therefore cross-calibrate and constrain the swell properties in both directions. Besides that, phase-difference spectra might help infer wave properties in areas where intensity spectra are contaminated by for example ice ridges (Ardhuin et al., 2017).

Figure 10 shows the elevations, the intensity and normalized elevation spectra computed from OceanSAR simulations of an ice surface that is exposed to swell waves in four directions. The peaks in the intensity spectra are expected at $(\cos\phi_w, \sin\phi_w)$ with $\phi_w$ the direction of the wave propagation. Note again that $\phi_w = 0°$ is the effective range direction of the bi-static system, which is in case of Harmony about $-16°$ from the cross-track direction. This is indeed true for the intensity spectra, which furthermore show the virtual absence of sensitivity to swell in the ground-range direction. Interpretation of the spectra is not straightforward due to the non-linear mapping from velocity gradients to intensity, which causes secondary harmonics (peaks) in the observed spectra. Additionally, the velocity variance of the surface causes signals at high wave numbers to dampen (Krogstad et al., 1994), which is of primary importance near the sea-ice edge where wind waves have not been fully dampened yet. Spectral transforms, taking into account both effects are required to infer wave amplitude or energy (Hasselmann and Hasselmann, 1991; Krogstad et al., 1994). At the moment of writing, bi-static transforms to convert ocean wave spectra into intensity spectra are being developed, which will be addressed in a separate paper.

The elevation spectra in figure 10 show the sensitivity to waves propagating in the direction of a monostatic equivalent between Harmony and Sentinel-1, which is directed about -16°, from the cross-track direction. Note that, our model considers a monostatic system, which slightly differs from the actual monostatic equivalent in the sense that the range and azimuth direction are not perpendicular to each other, but are rotated with respect to each other by a small angle. After normalization, the amplitude of the wave propagating in the receiver range direction ($\phi_w = 0°$) is close to the expected value of 2 meter. This, however, changes whenever $\phi_w$ is different, which is the consequence of the SAR image distortions as a consequence of a moving surface that change the retrieved elevation pattern. At larger azimuth angles, the amplitude reduces further, such that waves propagating in the flight direction are almost not detectable. In practice, this means that the inferred XTI height cannot directly be used to estimate wave height, but spectral transforms or geophysical model functions are required. The bi-static transforms to estimate wave height from phase information are yet to be developed.

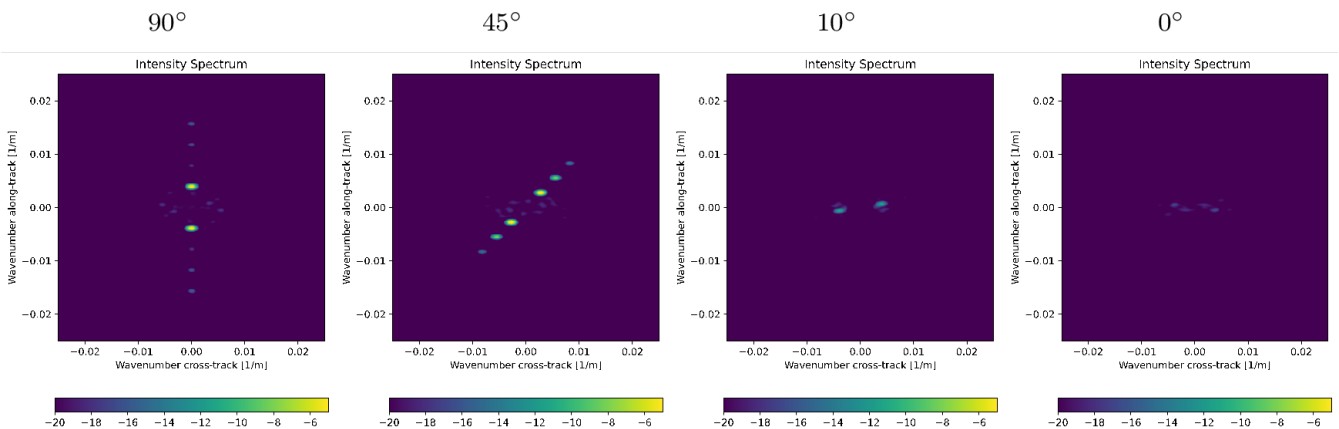

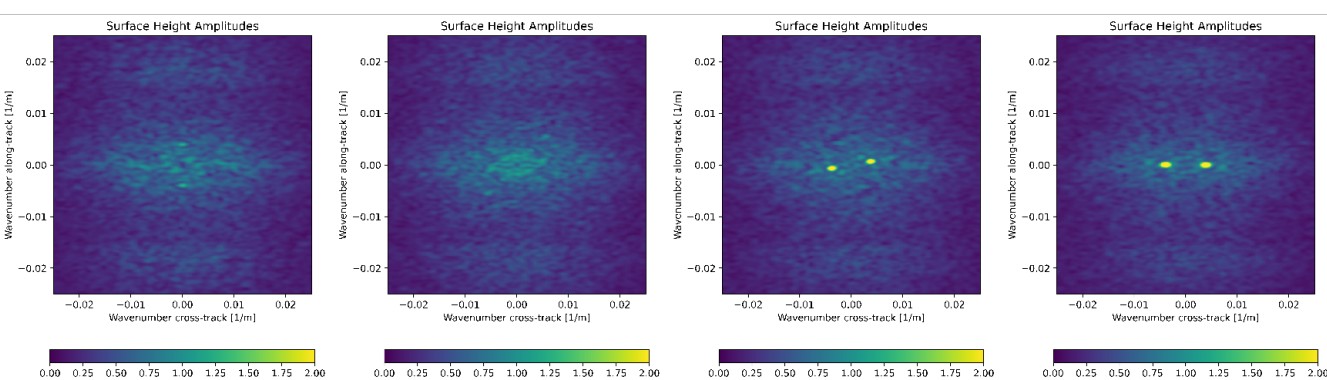

**Figure 10.** Observed intensity (log-scale) and elevation amplitude spectra (m) for 256-meter long, 2-meter high swell waves propagating in different directions. The 90°-direction corresponds to waves propagating in the azimuth direction.

Swell is detectable using SAR intensity spectra if the wave height is a few decimeters (Ardhuin et al., 2017), depending on the wave direction and sea-ice properties. The Stereo configuration of Harmony satellites provides better constraints for wave spectra in case waves are propagating near the range direction, but it also allows to cross-calibrate the spectra against each other. In XTI formation, once spectral transforms have been developed, two quasi-independent methods (height and intensity spectra) allow for the recovery of swell spectra in all directions plus again a cross-calibration of the swell-wave spectra derived from those methods. In both formations, Harmony data allows for a full two-dimensional ray-tracing of swell waves from a single pass until their dissipation to several decimeters height. This enables us to better constrain dissipation parameters when waves propagate under sea ice.

## 4    Conclusions

We presented the observation principle, several processing methods and the simulated performance of Harmony for sea ice. Harmony's formation with Sentinel-1D is reconfigurable and will operate in a stereo formation and a cross-track interferometry (XTI) formation. There are two main benefits of the bi-static stereo mode operation with respect to currently-operated monostatic missions, which are the abilities to vectorize the sea-ice drift estimates and to obtain instantaneous observations. The XTI mode has the benefit of instantly retrieving phase differences, so that decorrelation due to fast motion is minimal.

Harmony's stereo mode data will allow for the estimation of two-dimensional sea-ice drift fields and infer shear and divergence at floe boundaries. Both are required for the calibration of models and to determine local sea-ice properties. Due to the limited line-of-sight diversity of the constellation has the estimated cross-track velocity a lower uncertainty than the along-track velocity. Advanced post-processing strategies, like filtering and edge detection, are required to get the maximum out of the data. The performance is strongly enhanced by switching Sentinel-1D operations from the EW to the IW mode over sea ice. As the choice to operate Sentinel-1 in EW mode was driven by sea-ice community, we recommend to discuss the change of modes over at least parts of the sea-ice-covered regions within the sea-ice community.

Harmony also enables the estimation of two-dimensional swell-wave spectra and wave dissipation constants. It will not only enhance the understanding of wave propagation, but also improve the understanding of energy transfer to the ice, which could lead to events like breaking. In the stereo mode, Harmony will benefit from multiple lines-of-sight to remove the blind spot (i.e. swell propagating in the range direction) present in monostatic systems, and can use three observations from different angles to cross-calibrate the geophysical transfer model. In XTI mode, Harmony benefits from the instantaneous elevation retrieval, which is most sensitive to swell propagating in the receiver line-of-sight, while having two lines-of-sight for traditional velocity-bunching based spectra retrievals.

## Appendix A:  Snow-ice scattering model

Over sea ice Sentinel-1 transmits signals in a horizontal polarization and Harmony receives in both horizontal and vertical polarizations. Therefore only the two equations for HH and HV are required, which depend on the incoming incidence and

azimuth angle $(\theta_i, \phi_i)$ and the reflected angles $(\theta_r, \phi_r)$. Let $\mathbf{q_{i,r}}$ be the projections of the wave vectors on the horizontal plane for the incoming and received signals similarly to Komarov et al. (2014). The backscatter coefficient in HH for the air-snow interface is computed as

$$\sigma_{HH,air-snow} = \frac{k_0^4 |\Delta\epsilon_{as}|^2}{4\pi} |[1 + R_H(q_i)][1 + R_H(q_r)]|^2 \cos^2(\phi_r - \phi_i) K_s(\mathbf{q_r} - \mathbf{q_i}) \tag{A1}$$

and for the snow-ice interface as

$$\sigma_{HH,snow-ice} = \frac{k_0^4 |\Delta\epsilon_{si}|^2}{4\pi} |L_H(q_i) L_H(q_r)|^2 \cos^2(\phi_r - \phi_i) K_i(\mathbf{q_r} - \mathbf{q_i})). \tag{A2}$$

The cross-polarization terms HV are computed with

$$\sigma_{HV,air-snow} = \frac{k_0^4 |\Delta\epsilon_{as}|^2}{4\pi} |[1 + R_H(q_i)][1 - R_V(q_r)] \sin(\phi_r - \phi_i) \cos\theta_r|^2 K_s(\mathbf{q_r} - \mathbf{q_i})) \tag{A3}$$

and

$$\sigma_{HV,snow-ice} = \frac{k_0^4 |\Delta\epsilon_{si}|^2}{4\pi} |L_H(q_i) M_V(q_r)|^2 \sin^2(\phi_r - \phi_i) K_i(\mathbf{q_r} - \mathbf{q_i}). \tag{A4}$$

In the above equations $\Delta\epsilon_{as}$ and $\Delta\epsilon_{si}$ are the differences between the air and snow dielectric constants and the snow and ice dielectric constants, respectively. The spatial power spectal densities of the air-snow $K_s(\mathbf{q_r} - \mathbf{q_i})$ and the snow-ice $K_i(\mathbf{q_r} - \mathbf{q_i})$ interfaces are considered to be isotropic (i.e., with no dependence on the azimuth angle), and therefore we take them as (Komarov et al., 2015)

$$K_{s,i}(|\mathbf{q_r} - \mathbf{q_i}|) = \frac{2\pi L_{s,i}^2 \sigma_{s,i}^2}{(1 + |\mathbf{q_r} - \mathbf{q_i}|^2 L_{s,i}^2)^{1.5}}, \tag{A5}$$

where $L_{s,i}$ are the correlation lengths and $\sigma_{s,i}$ are the root-mean square (RMS) heights of the rough interfaces. The auto-correlation functions of both the air-snow and snow-ice interfaces are considered to be exponential. Using the scheme in the appendix of Komarov et al. (2015), the reflection coefficients $R_H(q_{i,r})$ and $R_V(q_{i,r})$ are computed, such that

$$R_H(q_{i,r}) = r_{d,H}^{0,1}(q_{i,r}) + \frac{t_{d,H}^{0,1}(q_{i,r}) t_{u,H}^{0,1}(q_{i,r}) r_{d,H}^{1,2}(q_{i,r}) u_1^2}{1 - r_{u,H}^{0,1}(q_{i,r}) r_{d,H}^{1,2}(q_{i,r}) u_1^2} \tag{A6}$$

and

$$R_V(q_{i,r}) = r_{d,V}^{0,1}(q_{i,r}) + \frac{t_{d,V}^{0,1}(q_{i,r}) t_{u,V}^{0,1}(q_{i,r}) r_{d,V}^{1,2}(q_{i,r}) u_1^2}{1 - r_{u,V}^{0,1}(q_{i,r}) r_{d,V}^{1,2}(q_{i,r}) u_1^2}, \tag{A7}$$

with $t, r$ the Fresnel transmission and reflection coefficients, $u, d$ indicating upward or downward, $H, V$ the polarization and $0, 1, 2$ the air, snow and ice layers, respectively. For example $t_{u,H}^{0,1}(q_i)$ is the upward Fresnel coefficient of the incoming signal for the air-snow interface for the horizontal polarization. The factor $u_1$ represents a phase change through the snow layer with thickness $\Delta z$ and is given by

$$u_1 = exp(iw_1 \Delta z), \tag{A8}$$

where $\epsilon_s$ is the dielectric constant of the snow layer and

$$495 \quad w_1 = k_0 \sqrt{\epsilon_s - \sin^2 \theta_i} \qquad \text{(A9)}$$

is the projection of the wave vector onto the z-axis in the snow layer. The two auxiliary variables $L_H$ and $M_V$ as given in the backscatter equation are given using the notations in the same scheme, such that

$$L_H(q_{i,r}) = \frac{w_0(q_{i,r})}{w_1(q_{i,r})} \frac{t_{u,H}^{0,1}(q_{i,r}) u_1}{1 + r_{d,H}^{1,2}(q_{i,r}) \cdot r_{d,H}^{0,1}(q_{i,r}) \cdot u_1^2} [1 + r_{d,H}^{1,2}(q_{i,r})] \qquad \text{(A10)}$$

and

$$500 \quad M_V(q_{i,r}) = \frac{w_0(q_{i,r})}{k_0} \frac{t_{u,V}^{0,1}(q_{i,r}) u_1}{1 + r_{d,V}^{1,2}(q_{i,r}) \cdot r_{d,V}^{0,1}(q_{i,r}) \cdot u_1^2} [1 - r_{d,V}^{1,2}(q_{i,r})]. \qquad \text{(A11)}$$

The term $w_0$ is the projection of the wave vector onto the z-axis in air, such that

$$w_0 = k_0 \cos \theta_i. \qquad \text{(A12)}$$

*Author contributions.* M.K., A.T., Y.L. and P.L. performed the computations. A.K. performed the neXtSIM sea-ice model runs. M.K. and P.L. wrote the manuscript. All authors contributed to the discussion and reviewed the manuscript.

*Competing interests.* Declarations of interest: none.

*Acknowledgements.* This activity was funded by the Dutch Space Office (NSO) in support of the Harmony Phase-0 studies (ref. NF-BINS1903).

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
