# Peer review of "Estimating instantaneous sea-ice dynamics from space using the bi-static radar measurements of Earth Explorer 10 candidate Harmony"

_The Cryosphere, 2020_

## Referee Comment (RC1) · Wolfgang Dierking (Referee) · 26 Nov 2020

Review: Estimating instantaneous sea ice dynamics from space using the bi-static radar measurements of Earth Explorer 10 candidate Harmony

by Marcel Kleinherenbrink and 8 co-authors

General comments:

Overall, the authors present a very thorough analysis regarding the potential of the

[Figure]

Harmony mission concept for retrieving parameters characterizing sea ice dynamics: short-term ice drift and deformation (here divergence and shear) as well as spectra of ocean waves travelling into sea ice at the marginal ice zone. Equations for calculating the sea ice drift vector from the "double" bistatic configuration are derived and explained; due to the lack of corresponding measurements, bistatic backscattering coefficients are calculated based on a theoretical model; and considering the noise-equivalent intensity, the "velocity noise" is estimated. In addition, a filtering scheme is suggested to reduce noise and preserve discontinuities in the drift velocity field. Finally, a possible methodology for the retrieval of wave spectra from ocean swell in sea ice is sketched. The paper is well written and structured and should definitely be published. Nevertheless, before publication some mostly easy-to-fix modifications are required to make the text easier to understand for readers who are not familiar with the details of this field of application.

Specific comments:

In addition, I think it is also necessary to sharpen the argument for the importance of Harmony in sea ice drift and deformation retrieval. A major argument in this paper for using data of the proposed Harmony mission is the availability of two-dimensional "instantaneous" sea ice drift vectors. What does "instantaneous" mean when linked with the Harmony mission? Why is the instantaneous drift important for understanding sea ice dynamics? In my opinion this requires a more detailed explanation.

The instantaneous ice drift is obtained from two measurements with a time offset $\Delta tb$ = BATI/2Vsat, equation 2 in the article. With 6.7 km/s (for Sentinel-1) and 300 km ATI-baseline, one obtains 22.4 s. This is a significant difference compared to the sub-seconds "instantaneous" movements from tandem ATI as described by Dammann et al. (The Cryosphere, 13, 1395–1408, 2019) or from Doppler-shift as proposed by Kræmer et al. (IEEE Transactions on Geoscience and Remote Sensing, Vol. 53, No. 12, December 2015) and should be mentioned.

The sea ice model simulations (section 2.1) were carried out with time steps of 200 s (hence larger by a factor of 10 compared to Harmony's temporal baseline). In order to assess how representative instantaneous snapshots are for characterizing sea ice drift and deformation, it would be useful to provide a typical range of time intervals over which sea ice drift can be assumed to be constant. Perhaps this is known from buoy measurements? It is understood that this is dependent on the temporal variability of the external forces and the internal "reaction time" of the ice. And how fast are ice velocities changing in break-up or closing events? It would strengthen the arguments for the need of the Harmony mission if these questions are addressed (e. g. in a discussion section), as well as including brief examples why and how the information about different temporal scales is actually used in the models.

It has also to be considered that there are large time gaps between single instantaneous drift measurements ("epochs" in Fig 7 of the article) – how large are these time intervals for an anticipated mission scenario, dependent on latitude? I assume that a direct comparison between model results and instantaneous observations will not be very useful considering the usual lack of knowledge about short-term spatial and temporal variations of the forces acting on the ice.

Page 16, lines 357-366: The retrieval of feature tracking and pattern matching is not restricted to image pairs acquired with a one day's temporal gap, this depends on the number of available satellites and the latitude. Nevertheless, it is true that the actual drift distance is underestimated at larger time gaps. But also Harmony cannot close this gap, since only short-term (20 s) snapshots of the drift field are available, with larger temporal gaps between single snapshots (and in this case, there is only one passive-active SAR formation, i.e. time gaps between short-term snapshots are the same as for the retrieval of the average drift from an image pair acquired with one satellite). The advantage I see is that we will get a much better picture of possible variations and temporal scaling of the drift velocity (also available from buoys) with high spatial resolution over a large area at a given time (not possible with buoys). Here, both

the traditional tracking methods and bistatic measurements provide important contributions. For the marginal ice zone, the retrieval of ice drift employing feature tracking and pattern matching is indeed often very difficult, this is hence another advantage of Harmony (but still the argument of the long time gaps between snapshots applies).

Technical comments:

(a) "Speckle tracking" requires coherency, and results in sub-pixel accuracies of (slow glacier) ice displacements. However, unfortunately we cannot use this technique for the retrieval of sea ice drift from two overlapping SAR images that were acquired with a time gap of several hours. In this case speckle is NOT correlated, and the methods used for estimating average displacement or drift vectors are either feature tracking or pattern matching (described, e.g. in A. A. Korosov, P. Rampal, "A combination of feature tracking and pattern matching with optimal parametrization for sea ice drift retrieval from SAR data", Remote Sensing 9, 258).

(b) I recommend to include the papers by Dammann et al. and Kræmer et al. mentioned above either in your introduction or in a discussion section and emphasize the difference between their and your "instantaneous" drift field.

(c) A table with numbers for baselines (ATI, XTI, critical) and other Harmony mission parameters would be useful.

(d) Bistatic SAR, "bistatic angle", "bistatic distance", along and across-track baselines for stereo and close-formation mode – since many readers will not be familiar with the concept of bistatic SAR, a drawing showing the major geometric elements used in your text would also be useful (extension of Fig. 1, or addition to it)?

(e) Page 5: At the bottom you refer to the previous section in which you describe orientation and geometry but on page 3 you mention that a detailed overview of Harmony's observation geometry will be discussed in a separate publication (line 80)? Please explain the angles in equation (3).

(f) Page 6: Equations under (5) are central for this paper. This may also include equation (7). In what mostly follows after equation (5), your focus is on the drift field (and not radar intensity). I think one should emphasize this even more strongly than done in the recent text.

(g) On page 7, line 170, you state that you can ignore baseline decorrelation in the Stereo formation because the phase centers are separated by only a few meters. Here you talk about cross-track decorrelation between each of the Harmony satellites and the Sentinel-1 satellite? What about the temporal baseline in XTI-(close-) formation?

(h) You assume that the volume decorrelation can be ignored. You can mention that formally this is not a problem: if the volume decorrelation is < 1, then the total decorrelation in Eq. (8) decreases further. Hence your results can also account for the effect of volume decorrelation, but you can't link its magnitude with specific ice properties (low salinity ice such as multi-year or brackish ice).

(i) Is the description of the Komarov model in such detail necessary, or would it be sufficient to refer to the two referred Komarov papers? If you think that it is necessary to include all equations in the paper you should move them to an appendix.

(j) What are the criteria for selecting a certain scale factor in Eq. (32)?

(k) Page 11, line 271: Edges are kept if they have more than 50 pixels connected together - what is the pixel size (is it 1km x 1km as mentioned one page earlier, line 251)? This means that discontinuities shorter than 50 km are ignored in the further analysis (determination of divergence and shear)? Please clarify and give a reason for this threshold.

(l) Page 12 line 309: Which ICESAR campaign are you referring to (any reference)? Line 327: does it make sense to consider a negative SNR?

(m) Page 13, Fig. 3, caption: These values were "taken" from Komarov et al. (2019) and Landy et al. (2019). ("agree" indicates somehow that these ice parameters are a

result of your calculations). Line 334: "A change of operational mode will therefore be beneficial for sea ice studies" – valid if you focus on the retrieval of instantaneous drift (and for a few other applications), but not necessarily valid in general. This should be noted.

(n) Page 14: the subswath edges can't be recognized in Fig. 5. Perhaps you should add a zoom-in.

(o) Page 15, line 355: It is ROSE-L, not L-ROSE

(p) Page 16: You talk about velocities in leads - do you mean pieces of ice in leads or the surface current of the water surface? In the marginal ice zone, ice floes may rotate more than in closed pack ice. The rotation causes also a phase shift. Any comments on that? Line 386-387: I disagree – we don't see single ice floes in Fig. 8 and 9 but separate regions with different drift velocity vectors.

(q) Fig. 7: the instantaneous velocities were calculated for a time step of 200 s? Figs. 8 and 9: please add the extent of the images in meters or provide pixel size.

(r) Page 19, lines 410-411, sentence: "This allows to infer swell properties in any direction as the gap is typically smaller than this." I don't understand the second part of the sentence, which gap is smaller than what?

(s) Page 20, line 423 "or" => of

(t) Just curiosity: you mention that the Harmony satellites shall be equipped with a TIR sensor- what are its potential applications? Dependent on the position of the target area of the TIR sensors, they could help to detect thin sea ice and open water leads (under cloud-free conditions).

---

## Referee Comment (RC2) · Leif Toudal Pedersen (Referee) · 17 Feb 2021

This is a nice and well written paper describing the potential capabilities of the EE-10 candidate Harmony.

I have a number of minor comments/suggestions:

In general many terms in equations are not properly defined, please correct in the entire document.

[Figure]

L7: speckle tracking -> feature tracking Speckle tracking is a well defined method of tracking mainly applied to relatively slow moving targets such as ice sheets and glaciers. The method is not applicable to sea ice with hours/days between observations, since the backscatter phase will totally decorrelate. Please change this in the entire document.

L15: will -> could

L27: Ricker et al, 2017 is hardly a proper reference for ICESat-2

L37: speckle tracking -> feature tracking

L37: while covering the both poles -> while not covering the poles

L41: Please provide a reference to the claim that instantaneous velocities are an order of magnitude larger than the daily averages in breakup events.

L122-3: A pan-arctic . . . - this sentence seems unfinished, please rephrase

L130-: Many terms in the equations are not defined. Please do so.

L138: Doppler -> Doppler shift

L144-45: Please argue how the distances will be kept equal to the accuracy required, in other words that this is a reasonable assumption.

L167: Please argue why it is reasonable to set the volume decorrelation to 1 (for MY-ice)

L168: difference -> differences

L220-230: These equations are valid for plane parallel layers with no internal scattering – is that a reasonable assumption?

L252: Please define PSD

L257: Negative values may me un-physical, but setting them to 0 may lead to a bias if

the below-zero value came about due to noise? Please discuss.

L279: Please provide a reference for the OceanSAR software package

L294: Please provide a reference to Bartlett's method.

L323: Please rephrase the first line of this sentence – it does not read well.

Figure 3: Some explanation titles in the 3 subplots would be helpful.

Figure 4: A legend inside the plots explaining the full and dotted lines would be good.

L335: speckle tracking -> feature tracking

L342: At the resolution and quality provided here, the subswaths are NOT 'clearly visible' in Fig 5.

L349: flows -> floes

L347-356: This argument is quantitatively somewhat flawed since there are other limiting factors to how much polar coverage S1D will deliver such as SAR duty-cycle and land application requirements. This should be stated/discussed.

L355: L-ROSE -> ROSE-L

L356: speckle tracking -> feature tracking

L359: speckle tracking -> feature tracking

L364: speckle tracking -> feature tracking

L371-72: This argument seems to assume instant response of the ice drift to wind. This is not what we have in reality. Please argue or modify/explain better what you mean or how this could be achieved.

L384: flows -> floes Figure 8: It might be more illustrative to have the same x-range on the 2 histograms for a better inter-comparison.

L398: to occur -> to be observed Figure 9: Please explain better why the larger shear are missed by the Harmony estimate

L454-55: Due to . . . - this statement needs rewriting.

L458-59: Please clarify that this statement is on your behalf and NOT on behalf of the sea-ice community at large. Other requirements such as more frequent coverage is better obtained using EW.

Some further considerations about the uncertainties in the results presented could be helpful.

---

## Author Comment (AC1) · 18 Mar 2021

**Author's Response to "Estimating instantaneous sea-ice dynamics from space using the bistatic radar measurements of Earth Explorer 10 candidate Harmony"**

Kleinherenbrink et al. (2020)

Dear Chris Derksen, Wolfgang Dierking and Leif Toudal Pedersen,

We would like to thank both reviewers for their thorough review of our manuscript. We have implemented almost all the recommended adjustments and updated the manuscript where more clarity is required.

This response consists of a rebuttal to reviewer 1, Wolfgang Dierking, a rebuttal to reviewer 2, Leif Toudal Pedersen, and at the end a list of additional adjustments.

Marcel on behalf of all authors

**1 Reviewer 1**

**General comments:**

**Overall, the authors present a very thorough analysis regarding the potential of the Harmony mission concept for retrieving parameters characterizing sea ice dynamics: short-term ice drift and deformation (here divergence and shear) as well as spectra of ocean waves travelling into sea ice at the marginal ice zone. Equations for calculating the sea ice drift vector from the "double" bistatic configuration are derived and explained; due to the lack of corresponding measurements, bistatic backscattering coefficients are calculated based on a theoretical model; and considering the noise-equivalent intensity, the "velocity noise" is estimated. In addition, a filtering scheme is suggested to reduce noise and preserve discontinuities in the drift velocity field. Finally, a possible methodology for the retrieval of wave spectra from ocean swell in sea ice is sketched. The paper is well written and structured and should definitely be published. Nevertheless, before publication some mostly easy-to-fix modifications are required to make the text easier to understand for readers who are not familiar with the details of this field of application.**

**Specific comments: In addition, I think it is also necessary to sharpen the argument for the importance of Harmony in sea ice drift and deformation retrieval. A major argument in this paper for using data of the proposed Harmony mission is the availability of two-dimensional"instantaneous" sea ice drift vectors. What does "instantaneous" mean when linked with the Harmony mission? Why is the instantaneous drift important for understanding sea-ice dynamics? In my opinion this requires a more detailed explanation.**
Instantaneous is in case of Harmony the average velocity in a fraction of a second. We have adjusted the text and indicated that the primary importance is that Harmony helps to improve model parametrizations, which will lead to more realistic sea-ice deformation and sea-ice evolution. Harmony's data will also allow for a validation of existing ice-motion products.

**The instantaneous ice drift is obtained from two measurements with a time offset $\Delta$tb= BATI/2Vsat, equation 2 in the article. With 6.7 km/s (for Sentinel-1) and 300 km ATI-baseline, one obtains 22.4 s. This is a significant difference compared to the sub-seconds "instantaneous" movements from tandem ATI as described by Dammann et al. (The Cryosphere, 13, 1395-1408, 2019) or from Doppler-shift as proposed by Kræmer et al. (IEEE Transactions on Geoscience and Remote Sensing, Vol. 53, No.12, December 2015) and should be mentioned.**
The references have been included. ATI is not performed between Sentinel-1 and Harmony. Harmony receives the signal at two phase-centers (antennas) onboard of a single satellite. The receiving phase-centers are separated by only ∼6 m. You can basically see this as an overpass of two monostatic systems separate by ∼3 m, i.e. their zero-Doppler is 3 m apart in the along-track direction. The time offset is therefore only a fraction of a second. We have clarified the text.

**The sea ice model simulations (section 2.1) were carried out with time steps of 200 s(hence larger by a factor of 10 compared to Harmony's temporal baseline). In order to assess how representative instantaneous snapshots are for characterizing sea ice drift and deformation, it would be useful to provide a typical range of time intervals over which sea ice drift can be assumed to be constant. Perhaps this is known from buoy measurements? It is understood that this is dependent on the temporal variability of the external forces and the internal "reaction time" of the ice. And how fast are ice velocities changing in break-up or closing events? It would strengthen the arguments for the need of the Harmony mission if these questions are addressed (e. g. in a discussion section), as well as including brief examples why and how the information about different temporal scales is actually used in the models.**
neXtSIM is a sea ice model with brittle sea ice rheology (more precisely: Brittle-Bingham-Maxwell at the moment). It simulates the process of brittle sea ice deformation, which is realized as elastic deformation, followed by a brittle failure and then viscous deformation. Judging by satellite and buoy observations of sea ice deformation and intermittency at 1 - 1000 km spatial scale and at 1e3 - 1e6 sec temporal scale, this process is very well represented by neXtSIM. (A recent study by Hutter et al., shows that elasto-visco-plastic

models are not capable of achieving similar results). Moreover, the model parametrization and discretization scheme allow to resolve the process of deformation at time steps which are larger than the time required for an elastic wave to propagate at a given spatial scale (i.e. timestep of 200 s for 10 km). Reducing timestep does not change dramatically the simulation results. We can therefore conclude that the spatial distribution of ice deformation simulated by neXtSIM is representative at the effective spatial resolution of Harmony observations.

Concerning the temporal resolution, we cannot draw such conclusion because observations of ice deformation at such combination of temporal/spatial scales are not available yet. One of the goals of the Harmony mission is to provide such observations for assessment and, if necessary, calibration of sea ice models and improvement of rheology.

**It has also to be considered that there are large time gaps between single instantaneous drift measurements ("epochs" in Fig 7 of the article) – how large are these time intervals for an anticipated mission scenario, dependent on latitude? I assume that a direct comparison between model results and instantaneous observations will not be very useful considering the usual lack of knowledge about short-term spatial and temporal variations of the forces acting on the ice.**
We agree with the reviewer that a direct comparison with a model will not be beneficial due to undersampling of the forces acting on the ice. However, Harmony will allow for a statistical comparison of sea-ice drift and deformation. We have also indicated in the text that improved model parametrization are the main benefit of the Harmony mission.

**Page 16, lines 357-366: The retrieval of feature tracking and pattern matching is not restricted to image pairs acquired with a one day's temporal gap, this depends on the number of available satellites and the latitude. Nevertheless, it is true that the actual drift distance is underestimated at larger time gaps. But also Harmony cannot close this gap, since only short-term (20 s) snapshots of the drift field are available, with larger temporal gaps between single snapshots (and in this case, there is only one passive-active SAR formation, i.e. time gaps between short-term snapshots are the same as for the retrieval of the average drift from an image pair acquired with one satellite). The advantage I see is that we will get a much better picture of possible variations and temporal scaling of the drift velocity (also available from buoys) with high spatial resolution over a large area at a given time (not possible with buoys). Here, both the traditional tracking methods and bistatic measurements provide important contributions. For the marginal ice zone, the retrieval of ice drift employing feature tracking and pattern matching is indeed often very difficult, this is hence another advantage of Harmony (but still the argument of the long time gaps between snapshots applies).**
We agree with the reviewer and clarified in the text that Harmony still suffers from temporal undersampling, but its wide swath puts instantaneous drift estimates from buoys in context. Therefore Harmony allows for better constraints on modelled ice-motion vectors.

**Technical comments:**

(a) "Speckle tracking" requires coherency, and results in sub-pixel accuracies of (slow glacier) ice displacements. However, unfortunately we cannot use this technique for the retrieval of sea ice drift from two overlapping SAR images that were acquired with a time gap of several hours. In this case speckle is NOT correlated, and the methods used for estimating average displacement or drift vectors are either feature tracking or pattern matching (described, e.g. in A. A. Korosov, P. Rampal, "A combination of feature tracking and pattern matching with optimal parametrization for sea ice drift retrieval from SAR data", Remote Sensing 9, 258).

Agreed, we have changed the term to feature tracking.

(b) I recommend to include the papers by Dammann et al. and Kraemer et al. mentioned above either in your introduction or in a discussion section and emphasize the difference between their and your "instantaneous" drift field.

We have referred to both papers and we have included instantaneous drifts in the conclusion.

(c) A table with numbers for baselines (ATI, XTI, critical) and other Harmony mission parameters would be useful.

We have included a list of mission parameters.

(d) Bistatic SAR, "bistatic angle", "bistatic distance", along and across-track baselines for stereo and close-formation mode – since many readers will not be familiar with the concept of bistatic SAR, a drawing showing the major geometric elements used in your text would also be useful (extension of Fig. 1, or addition to it)?

We have incorporated more information on the geometry in figure 1 to make the equations easier to understand.

(e) Page 5: At the bottom you refer to the previous section in which you describe orientation and geometry but on page 3 you mention that a detailed overview of Harmony's observation geometry will be discussed in a separate publication (line 80)? Please explain the angles in equation (3).

The angles are explained in the text and added to the figure showing the geometry.

(f) Page 6: Equations under (5) are central for this paper. This may also include equation (7). In what mostly follows after equation (5), your focus is on the drift field (and not radar intensity). I think one should emphasize this even more strongly than done in the recent text.

In the first paragraph of 'methods' we clarify that the focus is on the drift field, so the Doppler observations. This is repeated again near equation 5.

(g) On page 7, line 170, you state that you can ignore baseline decorrelation in the Stereo formation because the phase centers are separated by only a few meters. Here you talk about cross-track decorrelation between each of the Harmony satellites and the Sentinel-1 satellite? What about the temporal baseline in XTI-(close-) formation?

We have added a sentence describing the effects of baseline decorrelation in the XTI mode. In principle we can keep the along-track baseline small using a helix formation in Harmony's slanted geometry, so the primary source of decorrelation will come from the cross-track baseline.

**(h) You assume that the volume decorrelation can be ignored. You can mention that formally this is not a problem: if the volume decorrelation is < 1, then the total decorrelation in Eq. (8) decreases further. Hence your results can also account for the effect of volume decorrelation, but you can't link its magnitude with specific ice properties (low salinity ice such as multi-year or brackish ice).**
We have adjusted the text here to incorporate the comments of the reviewer, but note that the phase centers are only separated by a baseline of 6 m. Volume decorrelation is therefore small.

**(i) Is the description of the Komarov model in such detail necessary, or would it be sufficient to refer to the two referred Komarov papers? If you think that it is necessary to include all equations in the paper you should move them to an appendix.**
The lead author was struggling quite a bit with the implementation of the model, therefore we decided to put it in such that the replication becomes straightforward. We have moved the description to the appendix.

**(j) What are the criteria for selecting a certain scale factor in Eq. (32)?**
The scale factor is set to 1, but it basically implies stronger or weaker filtering when changed. Stronger filtering implies a smearing of the discontinuities at the benefit of reducing noise. So, depending on your application, you can increase or decrease the parameter 's'.

**(k) Page 11, line 271: Edges are kept if they have more than 50 pixels connected together - what is the pixel size (is it 1km x 1km as mentioned one page earlier, line251)? This means that discontinuities shorter than 50 km are ignored in the further analysis (determination of divergence and shear)? Please clarify and give a reason for this threshold.**
In figure 8 we use 2 km x 2 km multilook, the sentence with 1 km x 1 km is removed. The 50 pixels together refer to the dilated images, so the length will only be a fraction of these 50 pixels. We adjusted the text and clarified the above.

**(l) Page 12 line 309: Which ICESAR campaign are you referring to (any reference)?**
We have removed the reference to the ICESAR campaign as this data is currently not publicly available.

**Line 327: does it make sense to consider a negative SNR?**
Yes, because it refers to the backscatter/NESZ. After multilooking the noise in the drift estimates is reduced with the factor sqrt(N).

**(m) Page 13, Fig. 3, caption: These values were "taken" from Komarov et**

al. (2015) and Landy et al. (2019). ("agree" indicates somehow that these ice parameters are a result of your calculations).

We have adjusted this sentence. Besides, we adjusted the snow and ice dielectric values, snow thickness, and roughness parameters according to the Case Study 2 of Komarov et al. (2015) which corresponds to thick (1.2 m) winter snow-covered FYI. Updated dielectric constants of snow and sea ice include imaginary parts now to properly account for the presence of brine in snow and sea ice. The Landy et al. (2019) reference was removed as the dielectric values in this paper correspond to Ku-band and not C-band.

**Line 334: "A change of operational mode will therefore be beneficial for sea ice studies" – valid if you focus on the retrieval of instantaneous drift(and for a few other applications), but not necessarily valid in general. This should be noted.**

We have adjusted the sentence as the reviewer recommended.

**(n) Page 14: the subswath edges can't be recognized in Fig. 5. Perhaps you should add a zoom-in.**

We have adjusted the sentence. The subswath edges will be visible in the figure 8.

**(o) Page 15, line 355: It is ROSE-L, not L-ROSE**

Updated.

**(p) Page 16: You talk about velocities in leads - do you mean pieces of ice in leads or the surface current of the water surface? In the marginal ice zone, ice floes may rotate more than in closed pack ice. The rotation causes also a phase shift. Any comments on that?**

We refer to the surface current of the water. The fringes, caused by rotation, visible in Tandem-X will hardly be detectable with Harmony as the baseline is much shorter and the resolution used here is lower.

**Line 386-387: I disagree – we don't see single ice floes in Fig. 8 and 9 but separate regions with different drift velocity vectors.**

Agreed, we have adjusted the sentence.

**(q) Fig. 7: the instantaneous velocities were calculated for a time step of 200 s?**

Yes, and this is clarified in the caption.

**Figs.8 and 9: please add the extent of the images in meters or provide pixel size.**

We have adjusted the axes.

**(r) Page 19, lines 410-411, sentence: "This allows to infer swell properties in any direction as the gap is typically smaller than this." I don't understand the second part of the sentence, which gap is smaller than what?** The sentence was incorrect. We have removed the sentence and we clarified the description of the swell estimation.

**(s) Page 20, line 423 "or" => of**
Changed.

**(t) Just curiosity: you mention that the Harmony satellites shall be equipped with a TIR sensor- what are its potential applications? Dependent on the position of the target area of the TIR sensors, they could help to detect thin sea ice and open water leads(under cloud-free conditions.**
Both Harmony satellites will be equipped with a TIR instrument with sensors pointing towards in five along-track directions covering the swath. In Stereo-formation overlapping scenes between the TIRs of two cameras can be used to estimate cloud-top height and as they illuminate the same area at a slightly different time offset, also (horizontal) cloud velocity. When there are no clouds, it is possible to estimate temperature gradients at the surface. This will indeed likely help to identify thin sea ice or water. Note that the resolution will be O(500 m).

**2   Reviewer 2**

**This is a nice and well written paper describing the potential capabilities of the EE-10candidate Harmony. I have a number of minor comments/suggestions:**
**In general many terms in equations are not properly defined, please correct in the entire document.**
We made sure that all the terms in the equations are defined.

**L7: speckle tracking -> feature tracking Speckle tracking is a well defined method of tracking mainly applied to relatively slow moving targets such as ice sheets and glaciers. The method is not applicable to sea ice with hours/days between observations, since the backscatter phase will totally decorrelate. Please change this in the entire document.**
The term speckle tracking is indeed not correct and is changed to feature tracking throughout the document.

**L15: will -> couldL27: Ricker et al, 2017 is hardly a proper reference for ICESat-2**
Agreed, it only refers to CryoSat-2 studies over ice. We have added two references.

**L37: speckle tracking -> feature tracking**
Changed.

**L37: while covering the both poles -> while not covering the poles**
Updated.

**L41: Please provide a reference to the claim that instantaneous velocities are an order of magnitude larger than the daily averages in breakup events.**

We have adjusted the sentence. From virtual buoys tracked throughout a sequence of coastal radar images with 2 minutes temporal resolution (Karvonen et al. (2016)) it appears that instantaneous velocities can be at least 3-4 times larger than the daily averages in break up events.

**L122-3: A pan-arctic...- this sentence seems unfinished, please rephrase**
The sentence is rephrased.

**L130-: Many terms in the equations are not defined. Please do so.**
We double-checked that all the terms are defined.

**L138: Doppler -> Doppler shift**
Changed.

**L144-45: Please argue how the distances will be kept equal to the accuracy required, in other words that this is a reasonable assumption.**
We note now that the formation will vary over time and that more accurate estimated can be computed taking the exact geometry into account.

**L167: Please argue why it is reasonable to set the volume decorrelation to 1 (for MY-ice)**
Volume decorrelation depends on the baseline between the antennas, which is typically several hundreds of meters. In the case of Harmony this is only 6 meters, so volume decorrelation is virtually absent. We clarified this in the text.

**L168: difference -> differences**
Updated.

**L220-230: These equations are valid for plane parallel layers with no internal scattering– is that a reasonable assumption?**
The layers are assumed to be plane parallel and the dominant scattering is at the air-snow and snow-ice rough interfaces. We clarified in the text that this assumption is not always valid, but that the model is simply used to get reasonable estimates for the backscattering. The actual backscatter will change substantially geographically, so in reality you will see patches where the SNR is better or worse than the 'mean' used in this study.

**L252: Please define PSD**
Defined in the text. Power Spectral Density.

**L257: Negative values may me unphysical, but setting them to 0 may lead to a bias if the below-zero value came about due to noise? Please discuss.**
In the text we mention that this assumption leads to the elimination of some signals below or near the noise level, slightly biasing the results.

**L279: Please provide a reference for the OceanSAR software package**
A link to the OceanSAR software package has been added.

**L294: Please provide a reference to Bartlett's method.**
Reference added.

**L323: Please rephrase the first line of this sentence – it does not read well.**
Sentence rephrased.

**Figure 3: Some explanation titles in the 3 subplots would be helpful.**
We have added titles to both figures 3 and 4.

**Figure 4: A legend inside the plots explaining the full and dotted lines would be good.**
We have updated the legend.

**L335: speckle tracking -> feature tracking**
Updated.

**L342: At the resolution and quality provided here, the subswaths are NOT 'clearly visible' in Fig 5.**
Agreed, we have rephrased the sentence.

**L349: flows -> floes**
Updated.

**L347-356: This argument is quantitatively somewhat flawed since there are other limiting factors to how much polar coverage S1D will deliver such as SAR duty-cycle and land application requirements. This should be stated/discussed.**
We have indicated that the expected number of passes might be lower than the number reported here as it depends on the duty cycle of Sentinel-1.

**L355: L-ROSE -> ROSE-L**
Updated.

**L356: speckle tracking -> feature tracking**
Changed.

**L359: speckle tracking -> feature tracking**
Changed.

**L364: speckle tracking -> feature tracking**
Changed.

**L371-72: This argument seems to assume instant response of the ice drift to wind. This is not what we have in reality. Please argue or modify/explain better what you mean or how this could be achieved.**
We have weakened the sentence, such that the connection is a bit more subtle.

**L384: flows -> floes**
Updated.

**Figure 8: It might be more illustrative to have the same x-range on the 2 histograms for a better inter-comparison.**
We have updated this figure completely with a change of axes.

**L398: to occur -> to be observed**
Changed sentence.

**Figure 9: Please explain better why the larger shear are missed by the Harmony estimate**
We have added a short explanation in the caption and we adjusted the sentence "Secondly, the shear...". It is a matter of filtering.

**L454-55: Due to...- this statement needs rewriting.**
Sentence rephrased.

**L458-59: Please clarify that this statement is on your behalf and NOT on behalf of the sea-ice community at large. Other requirements such as more frequent coverage is better obtained using EW.**
We have changed the sentence, such that it is on behalf of us, and likely only for parts of the sea-ice-covered regions.

**Some further considerations about the uncertainties in the results presented could be helpful.**
We have added a short discussion in the text. The considered performance of the system is based on the preliminary design parameters, which can slightly change. The SNR depends on the backscatter and therefore the type and roughness parameters of sea ice. The performance is therefore expected to be better or worse in some areas. The proposed processing method might not be suitable in all parts of the sea-ice covered domain, depending on the size and shape of the ice floes.

**3   Additional changes**

Several typos have been removed.

Alexander Komarov was added as a co-author for validating results related to the scattering model and his contribution to the discussion. He also noted that the dielectric constants of snow and ice require an imaginary component. This leads to updated results for figures 3, 8 and 9.

Figures 8 and 9 have been updated to give them a better layout and axes units (km).

Two sentences in section 3.4 were incorrect in lines 414 and 431. This should not be in the direction of the receiver range, but in the direction of the mono-static equivalent. These sentences are adjusted. We have added several sentences to explain the bistatic observation of waves in more detail.

---

## Author Comment (AC2) · 18 Mar 2021

Dear Leif Toudal Pedersen,

Please find the response to your comments in the document attached to the response on reviewer 1.

Best regards,

Marcel on behalf of all authors

---

## Author Response (AR1)

Dear Chris Derksen,

Thanks for editing our manuscript. We have implemented all the changes mentioned in the author's response. Please find attached the adjusted manuscript and a version with changes highlighted.

With respect to the author's response, we have slightly adjusted the post-processing methods. As the adjusted di-electric constants gave a bit lower signal-to-noise ratio, we opted for a more robust implementation of the edge-detection algorithm.

Secondly, in the first author's response we stated that the distance between the two phase centers was 6 m, but we have used a distance of 9 m, which is up to date with the latest antenna designs.

Best regards,

Marcel Kleinherenbrink (on behalf of all author's)